# Rethinking Semi-Supervised Medical Image Segmentation: A Variance-Reduction Perspective

**Chenyu You**[1] **Weicheng Dai**[1] **Yifei Min**[1] **Fenglin Liu**[2] **David A. Clifton**[2]

**S. Kevin Zhou**[3] **Lawrence Staib**[1] **James S. Duncan**[1]

[1]Yale University  [2]University of Oxford  [3]University of Science and Technology of China

## Abstract

For medical image segmentation, contrastive learning is the dominant practice to improve the quality of visual representations by contrasting semantically similar and dissimilar pairs of samples. This is enabled by the observation that without accessing ground truth labels, negative examples with truly dissimilar anatomical features, if sampled, can significantly improve the performance. In reality, however, these samples may come from similar anatomical regions and the models may struggle to distinguish the minority tail-class samples, making the tail classes more prone to misclassification, both of which typically lead to model collapse. In this paper, we propose `ARCO`, a semi-supervised contrastive learning (CL) framework with stratified group theory for medical image segmentation. In particular, we first propose building `ARCO` through the concept of variance-reduced estimation and show that certain variance-reduction techniques are particularly beneficial in pixel/voxel-level segmentation tasks with extremely limited labels. Furthermore, we theoretically prove these sampling techniques are universal in variance reduction. Finally, we experimentally validate our approaches on eight benchmarks, *i.e.*, five 2D/3D medical and three semantic segmentation datasets, with different label settings, and our methods consistently outperform state-of-the-art semi-supervised methods. Additionally, we augment the CL frameworks with these sampling techniques and demonstrate significant gains over previous methods. We believe our work is an important step towards semi-supervised medical image segmentation by quantifying the limitation of current self-supervision objectives for accomplishing such challenging safety-critical tasks. [1]

## 1 Introduction

Model robustness and label efficiency are two highly desirable perspectives when it comes to building reliable medical segmentation models. In the context of medical image analysis, a model is said to be robust if (1) it has a high segmentation quality with only using extremely limited labels in long-tailed medical data; (2) and fast convergence speed [1, 2, 3]. The success of traditional supervised learning depends on training deep networks on a large amount of labeled data, but this improved segmentation/model robustness often comes at the cost of annotations and clinical expertise [4, 5, 6]. Therefore, it is difficult to adopt these models in real-world clinical applications.

Recently, a significant amount of research efforts [7, 8, 9, 10] have resorted to unsupervised or semi-supervised learning techniques for improving the segmentation robustness. One of the most effective methods is contrastive learning (CL) [11, 12, 13, 14]. It aims to learn useful representations

---

[1]Codes are available on here.

by contrasting semantically similar (positive) and dissimilar (negative) pairs of data points sampled from the massive unlabeled data. These methods fit particularly well with real-world clinical scenarios as we assume only access to a large amount of unlabelled data coupled with extremely limited labels. However, pixel-level contrastive learning with medical image segmentation is quite impractical since sampling all pixels can be extremely time-consuming and computationally expensive [15]. Fortunately, recent studies [16, 17] provide a remedy by leveraging the popular strategy of bootstrapping, which first actively samples a sparse set of pixel-level representation (queries), and then optimize the contrastive objective by pulling them to be close to the class mean averaged across all representations in this class (positive keys), and simultaneously pushing apart those representations from other class (negative keys). The demonstrated imbalancedness and diversity across various medical image datasets, as echoed in [18], show the positive sign of utilizing the massive unlabeled data with extremely limited annotations while maintaining the impressive segmentation performance compared to supervised counterparts. Meanwhile, it can lead to substantial memory/computation reduction when using pixel-level contrastive learning framework for medical image segmentation.

Nevertheless, in practical clinical settings, the deployed machine learning models often ask for strong robustness, which is far beyond the scope of segmentation quality for such challenging safety-critical scenarios. This leads to a more challenging requirement, which demands the models to be more robust to the *collapse* problems whereby all representations collapse into constant features [14, 13, 19] or only span a lower-dimensional subspace [20, 21, 22, 23], as one main cause of such fragility could be attributed to the non-smooth feature space near samples [24, 25] (*i.e.*, random sampling can result in large feature variations and even annotation information alter). Thus, it is a new perspective: *how to sample most informative pixels/voxels towards improving variance reduction in training semi-supervised contrastive learning models*. This inspires us to propose a new hypothesis of semi-supervised CL. Specifically, when directly baking in variance-reduction sampling into semi-supervised CL frameworks for medical image segmentation, the models can further push toward state-of-the-art segmentation robustness and label efficiency.

In this paper, we present ARCO, a semi-supervised str**A**tified g**R**oup **Co**ntrastive learning framework with two perspectives (*i.e.*, **segmentation/model robustness** and **label efficiency**), and with the aid of variance-reduction estimation, realize two practical solutions – Stratified Group (SG) and Stratified-Antithetic Group (SAG) – for selecting the most semantically informative pixels. ARCO is a group-based sampling method that builds a set of pixel groups and then proportionally samples from each group with respect to the class distribution. The **main idea** of our approach is via *first partitioning the image with respect to different classes into grids with the same size, and then sampling, within the same grid, pixels semantically close to each other with high probability, with minimal additional memory footprint*.

Subsequently, we show that baking ARCO into contrastive pre-training (*i.e.*, MONA [17]) provides an efficient pixel-wise contrastive learning paradigm to train deep networks that perform well in long-tailed medical data. ARCO is easy to implement, being built on top of off-the-shelf pixel-level contrastive learning framework [13, 14, 26, 27, 28], and consistently improve overall segmentation quality across all label ratios and datasets (*i.e.*, five 2D/3D medical and three semantic datasets).

Our theoretical analysis shows that, ARCO is more label efficient, providing practical means for computing the gradient estimator with improved variance reduction. Empirically, our approach achieves competitive results across eight 2D/3D medical and semantic segmentation benchmarks. Our proposed framework has several theoretical and practical contributions:

- We propose ARCO, a new CL framework based on stratified group theory to improve the label efficiency and model robustness trade-off in CL for medical image segmentation. We show that incorporating ARCO coupled with two special sampling methods, Stratified Group and Stratified-Antithetic Group, into the models provides an efficient learning paradigm to train deep networks that perform well in those long-tail clinical scenarios.

- To our best knowledge, we are the **first work** to show the benefit of certain variance-reduction techniques in CL for medical image segmentation. We demonstrate the unexplored advantage of the refined gradient estimator in handling long-tailed medical image data.

- We conduct extensive experiments to validate the effectiveness of our proposed method using a variety of datasets, network architectures, and different label ratios. For segmentation robustness/accuracy, we show that our proposed method by demonstrating superior segmentation accuracy (up to 11.08% absolute improvements in Dice). For label efficiency, our

method trained with different labeled ratios – consistently achieves competitive performance improvements across all eight 2D/3D medical and semantic segmentation benchmarks.

- Theoretical analysis of `ARCO` shows improved variance reduction with optimization guarantee. We further demonstrate the intriguing property of `ARCO` across the different pixel-level contrastive learning frameworks.

## 2 Related work

**Medical Image Segmentation.** Contemporary medical image segmentation approaches typically build upon fully convolutional networks (FCN) [29] or UNet [30], which formulates the task as a dense classification problem. In general, current medical image segmentation methods can be cast into two sets: network design and optimization strategy. One is to optimize segmentation network design for improving feature representations through dilated/atrous/deformable convolutions [31, 32, 33], pyramid pooling [34, 35, 36], and attention mechanisms [37, 38, 39]. Most recent works [40, 41, 6] reformulates the task as a sequence-to-sequence prediction task by using the vision transformer (ViT) architecture [42, 43]. The other is to improve optimization strategies, by designing loss function to better address class imbalance [44] or refining uncertain pixels from high-frequency regions improving the segmentation quality [45, 46, 47, 48, 49]. In contrast, we take a leap further to a more practical clinical scenario by leveraging the massive unlabeled data with extremely limited labels in the learning stage. Moreover, we focus on building *model-agnostic*, label-efficiency framework to improve segmentation quality by providing additional supervision on the most confusing pixels for each class. In this work, we question how medical segmentation models behave under such imbalanced class distributions and whether they can perform well in those challenging scenarios through sampling methods.

**Semi-Supervised Learning (SSL).** SSL aims to train models with a combination of labeled, weakly-labeled and unlabelled data. In recent years, there has been a surge of work on semi-supervised medical segmentation [8, 9, 50, 48, 16, 51, 52, 17, 10, 53, 54], which makes it hard to present a complete overview here. We therefore only outline some key milestones related to this study. In general, it can be roughly categorized into two groups: (1) Consistency regularization was first proposed by [55], which aims to impose consistency corresponding to different perturbations into the training, such as consistency regularization [56, 57], pi-model [58], and mean-teacher [59, 60]. (2) Self-training was initially proposed in [61], which aims at using a model's predictions to obtain noisy pseudo-labels for performance boosts with minimal human labor, such as pseudo-labeling [7, 62], model uncertainty [8, 63], confidence estimation [64, 65, 66], and noisy student [67]. These methods usually lead to competitive performance but fail to prevent *collapse* due to class imbalanceness. In this work, we focus on semi-supervised medical segmentation with extremely limited labels since the medical image data is extremely diverse and often long-tail distributed over anatomical classes. We speculate that a good medical segmentation model is expected to distinguish the minority tail-class samples and hence achieve better performance under additional supervision on hard pixels.

**Contrastive Self-Supervised Learning.** Self-supervised representation learning is a subclass of unsupervised learning, but with the critical distinction that it incorporates "inherent" supervision from the input data itself [68]. The primary aim of self-supervised representation learning is to enable the model to learn the most useful representations from the large amount of unlabelled data for various downstream tasks. Self-supervised learning typically relies on pretext tasks, including predictive [69, 70, 71], contextual [72, 73], and generative [74] or reconstructive [75] tasks.

Among them, contrastive learning is considered as a popular approach for self-supervised representation learning by pulling the representations of similar instances closer and representations of dissimilar instances further apart in the learned feature space [11, 12, 13, 14]. The past five years have seen tremendous progress related to CL in medical image segmentation [50, 23, 76, 48, 16, 17, 77], and it becomes increasingly important to improve representation in label-scarcity scenarios. The key idea in CL [11, 12, 13, 14] is to learn representations from unlabeled data that obey similarity constraints by pulling augmented views of the same samples closer in a representation space, and pushing apart augmented views of different samples. This is typically achieved by encoding a view of a data into a single global feature vector. However, the *global representation* is sufficient for simple tasks like image classification, but does not necessarily achieve decent performance, especially for more challenging dense prediction tasks. On the other hand, several works on *dense contrastive*

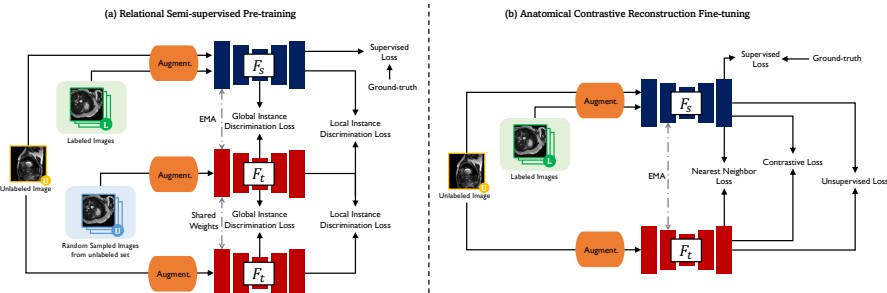

Figure 1: **Pipeline overview.** Our semi-supervised segmentation model $F$ takes a 2D/3D medical image $x$ as input and outputs the segmentation map and the representation map. We leverage a simplification of MONA pipeline [17] which is composed of two stages: (1) relational semi-supervised pre-training: on labeled data, the student network is trained by the ground-truth labels with the supervised loss $\mathcal{L}_{\text{sup}}$; while on unlabeled data, the student network takes the *augmened* and *mined* embeddings from the EMA teacher for instance discrimination $\mathcal{L}_{\text{inst}}$ in the global and local manner, (2) anatomical contrastive reconstruction fine-tuning: on labeled data, the student network is trained by the ground-truth labels with the supervised loss $\mathcal{L}_{\text{sup}}$; while on unlabeled data, the student network takes the representation maps and pseudo labels from the EMA teacher to give more importance to tail class $\mathcal{L}_{\text{contrast}}$, exploit the inter-instance relationship $\mathcal{L}_{\text{nn}}$, and compute unsupervised loss $\mathcal{L}_{\text{unsup}}$. See Appendix M for details of the visualization loss landscapes.

*learning* [50, 23], aim at providing additional supervision to capturing intrinsic spatial structure and fine-grained anatomical correspondence, while these methods may suffer from *class imbalance* issues. Particularly, very recent work [16, 17] for the first time demonstrates the imbalancedness phenomenon can be mitigated by performing contrastive learning yet lacking stability. By contrast, a key motivation of our work is to bridge the connection between model robustness and label efficiency, which we believe is an important and under-explored area. We hence focus on variance-reduced estimation in medical image segmentation, and show that certain variance-reduction techniques can help provide more efficient approaches or alternative solutions for handling *collapse* issues, and improving model robustness in terms of accuracy and stability. To the best of our knowledge, we are the first to provide a theoretical guarantee of robustness by using certain variance-reduction techniques.

## 3  Methodology

In this section we set-up our semi-supervised medical segmentation problem, introduce key definitions and notations and formulate an approach to incorporate stratified group theory. Then, we discuss how our proposed ARCO can directly bake in two perspectives into deep neural networks: (1) **model robustness**, and (2) **label efficiency**.

### 3.1  Preliminaries and setup

**Problem Definition.** In this paper, we consider the multi-class medical image segmentation problem. Specifically, given a medical image dataset $(\mathcal{X}, \mathcal{Y})$, we wish to automatically learn a segmentator, which assigns each pixel to their corresponding $K$-class segmentation labels. Let us denote $\mathbf{x}$ as the input sample of the *student* and *teacher* networks $F(\cdot)$ [2], consisting of an encoder $E$ and a decoder $D$, and $F$ is parameterized by weights $\theta_s$ and $\theta_t$.

**Background.** Contrastive learning aims to learn effective representations by pulling semantically close neighbors together and pushing apart other non-neighbors [11]. Among various popular contrastive learning frameworks, MONA [17] is easy-to-implement while yielding the state-of-the-art performance for semi-supervised medical image segmentation so far. The main idea of MONA is to discover diverse views (*i.e.*, augmented/mined views) whose anatomical feature responses are *homogeneous* within the same or different occurrences of the *same class type*, while at the same time being *distinctive* for *different class types*.

---

[2]The student and teacher networks both adopt the 2D UNet [30] or 3D VNet [78] architectures.

Hereinafter, we are interested in showing that certain variance-reduction techniques coupled with CL frameworks are particularly beneficial in long-tail pixel/voxel-level segmentation tasks with extremely limited labels. We hence build our `ARCO` as a simplification of the `MONA` pipeline [17], without additional complex augmentation strategies, for deriving the **model robustness** and **label efficiency** properties of our medical segmentation model. Figure 1 overviews the high-level workflow of the proposed `ARCO` framework. Training `ARCO` involves a two-phase training procedure: (1) relational semi-supervised pre-training, and (2) anatomical contrastive fine-tuning. To make the discussion self-contained, we defer the full details of `ARCO` to the appendix E.

## 3.2 Motivation and Challenges

Intuitively, the contrastive loss will learn generalizable, balanced and diverse representations for downstream medical segmentation tasks if the positive and negative pairs correspond to the desired latent anatomical classes [50, 16, 17]. Yet, one critical constraint in real-world clinical scenarios is severe *memory bottlenecks* [15, 16]. To address this issue, current pixel-level CL approaches [16, 17] for high-resolution medical images devise their aggregation rules by *unitary simulators*, *i.e.*, *Naïve Sampling* (NS), that determines the empirical estimate from all available pixels. Despite the blessing of large learning capacity, such aggregation rules are *unreliable* "black boxes". It is never well understood which rule existing CL models should use for improved **model robustness** and **label efficiency**; nor is it easy to compare different models and assess the model performance. Moreover, unitary simulators, especially naïve sampling, often incur high variances and fail to identify semantically similar pixels [24], limiting CL stability. As demonstrated in Figure 3, regions of similar anatomical features should be grouped together in the original medical images, resulting in corresponding plateau regions in the visualization of the loss landscape. This is consistent with the observations uncovered by the recent empirical findings [79, 80].

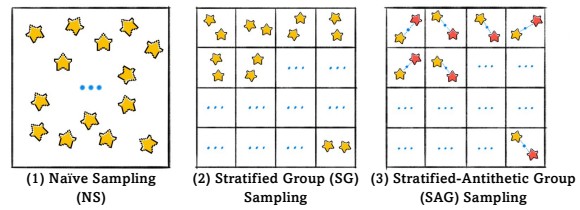

(1) Naïve Sampling (NS)    (2) Stratified Group (SG) Sampling    (3) Stratified-Antithetic Group (SAG) Sampling

Figure 2: Overview of three sampling methods. (1) Naïve Sampling, (2) Stratified Group Sampling, and (3) Stratified-Antithetic Group Sampling.

If we take a unified mathematical perspective, the execution of simulation can be represented either through an *adaptive* rule, or by a *unitary* simulation. To tackle the two critical issues, we look back at adaptive rules. We hence propose two straightforward yet effective techniques – Stratified Group (SG) and Stratified-Antithetic Group (SAG) – to mitigate the undesirable high-variance limitation, and turn to the following idea of sampling the most representative pixels from groups of semantically similar pixels. In particular, our proposed solution is based on stratified group simulation to adaptively characterize anatomical regions found on different medical images. This characterization is succinct, and regions with the same anatomical properties within different medical images are identifiable. *In practice, we first partition the image with respect to different classes into grids with the same size, and then sampling, within the same grid, the pixels semantically close to each other with high probability, with minimal additional memory footprint (Figure 2).*

In what follows, we will theoretically demonstrate the important properties of such techniques (*i.e.*, SG and SAG), especially in reduced variance and unbiasedness. Here the reduced variance implies more robust gradient estimates in the backpropagation, and leads to faster and stabler training in theory, as corroborated by our experiments (Section 4). Empirically, we will demonstrate many practical benefits of reduced variances including improved model robustness, *i.e.*, faster convergence and better segmentation quality, through mitigating the *collapse* issue.

## 3.3 Stratified Group Sampling

To be consistent with the previous notation, we denote an arbitrary image from the given medical image dataset as $\mathbf{x} \in \mathcal{X}$, and $\mathcal{P}$ as the set of pixels. For arbitrary function $h : \mathcal{X} \times \mathcal{P} \to \mathbb{R}$, we define

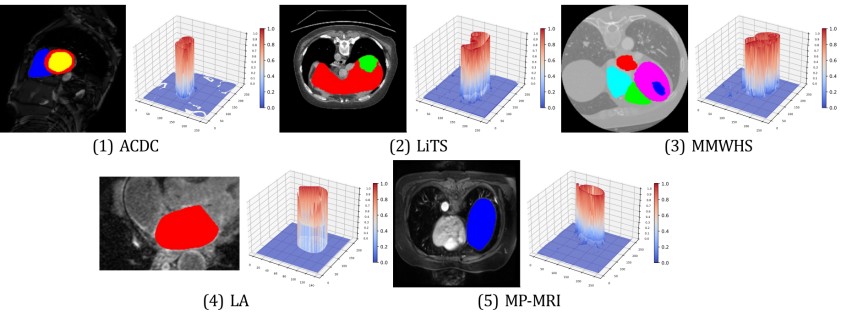

Figure 3: Loss landscape visualization of pixel-wise contrastive loss $\mathcal{L}_{\text{contrast}}$ with `ARCO-SG`. Loss plots are generated with same original images randomly chosen from ACDC [81], LiTS [82], MMWHS [83], LA [84], and MP-MRI, respectively. $z$-axis denotes the loss value at each pixel. For each example of the five benchmarks, the left subplot indicates that similar anatomical features are grouped together in the original medical images, as shown by different anatomical regions in different colors.

the aggregation function $H$[3] as:

$$H(\mathbf{x}) = \frac{1}{|\mathcal{P}|} \sum_{p \in \mathcal{P}} h(\mathbf{x}; p). \tag{3.1}$$

As a large cardinality of $\mathcal{P}$ prevents efficient direct computation of $H$, an immediate approach is to compute $H(\mathbf{x})$ by first sampling a subset of pixels $\mathcal{D} \subseteq \mathcal{P}$ according to certain sampling strategy, and then computing $\widehat{H}(\mathbf{x}; \mathcal{D}) = \sum_{p \in \mathcal{D}} h(\mathbf{x}; p)/|\mathcal{D}|$. SG sampling achieves this by first decomposing the pixels into $M$ disjoint groups $\mathcal{P}_m$ satisfying $\cup_{m=1}^{M} \mathcal{P}_m = \mathcal{P}$, and then sampling $\mathcal{D}_m \subseteq \mathcal{P}_m$ so that $\mathcal{D} = \cup_{m=1}^{M} \mathcal{D}_m$. The SG sampling can then be written as:

$$\widehat{H}_{\text{SG}}(\mathbf{x}; \mathcal{D}) = \frac{1}{M} \sum_{m=1}^{M} \frac{1}{|\mathcal{D}_m|} \sum_{p \in \mathcal{D}_m} h(\mathbf{x}; p).$$

SAG, built upon SG, adopts a similar form, except for an additionally enforced symmetry on $\mathcal{D}_m$: $\forall\, m,\, \exists\, c_m \in \mathcal{P}_m$, such that for any $p \in \mathcal{D}_m$,

$$c_m - p = p' - c_m, \quad \text{for some } p' \in \mathcal{D}_m.$$

Here $c_m$ denotes the center of the group $\mathcal{P}_m$[4]. The implementation of SG and SAG involves two steps: (1) to create groups $\{\mathcal{P}_m\}_{m=1}^{M}$, and (2) to generate each $\mathcal{D}_m \subseteq \mathcal{P}_m$. For the latter, we consider independent sampling within and between groups, *i.e.*, $\mathcal{D}_m \perp\!\!\!\perp \mathcal{D}_{m'}$ for $m \neq m'$, and $p \perp\!\!\!\perp p' \,\forall\, p, p' \in \mathcal{D}_m$, where the variance of SG sampling is as follows.

**Lemma 3.1.** *Suppose in SG sampling, for each $m$, $\mathcal{D}_m$ is sampled from $\mathcal{P}_m$ with sampling variance $\sigma_m^2$ and sample size $|\mathcal{D}_m| = n_m$. Then the variance satisfies $\text{Var}[\widehat{H}_{\text{SG}}] = \sum_{m=1}^{M} \sigma_m^2 n_m/n$, and SAG with the same sample size satisfies $\text{Var}[\widehat{H}_{\text{SAG}}] \leq 2\,\text{Var}[\widehat{H}_{\text{SG}}]$.*

To ensure the unbiasedness property, we adopt the setting of proportional group sizes [85, 86], *i.e.*, $|\mathcal{D}_m| \propto |\mathcal{P}_m|$ for all $m$. It turns out that such setting also enjoys the variance-reduction property.

**Theorem 3.2** (Unbiasedness and Variance of SG). *SG with proportional group sizes is unbiased, and has a variance no larger than that of NS. That is: $\mathbb{E}[\widehat{H}_{\text{SG}}(\mathbf{x})] = H(\mathbf{x})$, and*

$$\text{Var}[\widehat{H}_{\text{SG}}] = \text{Var}[\widehat{H}_{\text{NS}}] - \frac{1}{n} \sum_{m=1}^{M} \left( \mathbb{E}_{p \overset{\text{unif.}}{\sim} \mathcal{P}_m}[h(\mathbf{x}; p)] - \mathbb{E}_{p \overset{\text{unif.}}{\sim} \mathcal{P}}[h(\mathbf{x}; p)] \right)^2.$$

The last term is the intra-group variance, which captures the discrepancy between the pixel groups $\{\mathcal{P}_m\}_{m=1}^{M}$. Theorem 3.2 guarantees that the variance of SG is no larger than that of NS, and SG

---

[3]The pixel-level contrastive loss $\mathcal{L}_{\text{contrast}}$ is an example of an aggregation function (up to normalizing constant) according to Eqn. (E.2).

[4]The choice of $c_m$ is flexible. For example, if the convex hull of the pixels in $\mathcal{P}_m$ form a circle, then $c_m$ can be taken as the geometric center.

has strictly less variance than NS as long as all the pixel groups do not share an equal mean over $h(\mathbf{x}; p)$, which is almost-sure in medical images (See Figure 3). For SAG, Lemma 3.1 guarantees its variance is of the same magnitude as that of SG, and at worst differs by a factor of 2. Since the pixel/voxel-level contrastive loss $\mathcal{L}_{\text{contrast}}$ is an aggregation function over pixels by definition (E.2), it benefits from the variance-deduction property of SG/SAG. In Section 4.1, we will see that such variance reduction allows ARCO to achieve better segmentation accuracy, especially along the boundary of the anatomical regions (Figure 4).

**Training Convergence.** We further demonstrate the benefit of variance reduction estimation in terms of training stability. Specifically, leveraging techniques from standard optimization theory [87, 88, 89], we can show that variance-reduced gradient estimator through SG sampling leads to faster training convergence. Suppose we have a loss function $\mathcal{L}(\theta)$ with the model parameter $\theta$, and use stochastic gradient descent (SGD) as the optimizer. A gradient estimate $g(\theta) \approx \nabla \mathcal{L}(\theta)$ is computed at each iteration. It is well-known that the convergence of SGD depends on the quality of the estimate $g(\theta)$ [87]. Specifically, we make the common assumptions that the loss function is smooth and the gradient estimate has bounded variance (More details in Appendix A.3), which can be formulated as below:

$$\|\nabla \mathcal{L}(\theta) - \nabla \mathcal{L}(\theta')\|_2 \leq L(\|\theta - \theta'\|_2), \quad \mathbb{E}\left[\|g(\theta) - \nabla \mathcal{L}(\theta)\|^2\right] \leq \sigma_g^2.$$

Under these two assumptions, the average expected gradient norm of the learned parameter satisfies the following:

$$\frac{1}{T} \sum_{t=1}^{T} \mathbb{E}\left[\|\nabla \mathcal{L}(\theta_t)\|_2^2\right] \leq C\left(\frac{1}{T} + \frac{\sigma_g}{\sqrt{T}}\right).$$

For general non-convex loss function, the above implies convergence to some local minimum. Importantly, the slow rate $(\sigma_g/\sqrt{T})$ depends on standard deviation $\sigma_g$, indicating a faster convergence can indeed be achieved with a more accurate gradient estimate. This indicates our proposed sampling techniques demonstrate universality in variance reduction, as they can be applied to a wide range of scenarios that involve pixel/voxel-level sampling (See Appendix A.3). In Figure 5, we observe that using SG enables faster loss decay with smaller error bar, showing that it outperforms other methods in both convergence speed and stability. See Section 4.2 and Appendix A.3 for more details.

# 4 Experiments

In this section, we present experimental results to validate our proposed methods across various datasets and different label ratios in Appendix B. We use 2D `UNet` [30] or 3D `VNet` [78] as our backbones. Further implementation details are discussed in Appendix C. [5]

## 4.1 Main Results

In this subsection, we first examine whether our proposed ARCO can generalize well across various datasets and label ratios. Then, we investigate to what extent ARCO coupled with two samplers can realize two essential properties: (1) **model robustness**; and (2) **label efficiency**. The quantitative results for all the compared methods on eight popular datasets: (1) Medical image segmentation tasks: three 2D benchmarks (*i.e.*, ACDC [81], LiTS [82], MMWHS [83]), two 3D benchmarks (*i.e.*, LA [84], in-house MP-MRI) under various label ratios (*i.e.*, 1%, 5%, 10%) are collected in Table 1, Table 5 (Appendix G), Table 6 (Appendix H), and Table 9 (Appendix I and J), respectively; (2) General computer vision tasks: To further validate the effectiveness, we experiment on three popular segmentation benchmarks (*i.e.*, Cityscapes [97], Pascal VOC 2012 [98], indoor scene segmentation dataset – SUN RGB-D [99]) in the semi-supervised full-label settings. We follow the identical setting [100] to sample labelled images to ensure that every class appears sufficiently in our three datasets, (*i.e.*, CityScapes, Pascal VOC, and SUN RGB-D). The results are collected in Appendix Section K. Several consistent observations can be drawn from these extensive evaluations with eighteen segmentation networks.

❶ **Superior Performance Across Datasets.** We demonstrate that ARCO achieves superior performance across all datasets and label ratios. In specific, our experiments consider three 2D benchmarks

---

Table 1: Quantitative comparisons (DSC[%] ↑ / ASD[voxel] ↓) across the three labeled ratio settings (1%, 5%, 10%) on the ACDC benchmark. All experiments are conducted as [30, 90, 91, 92, 93, 8, 94, 9, 57, 95, 62, 50, 59, 96, 53, 16, 17] in the identical setting for fair comparisons. Best and second-best results are coloured **blue** and red, respectively. UNet-F (fully-supervided) and UNet-L (semi-supervided) are considered as the upper bound and the lower bound for the performance comparison. Note that, Right Ventricle → RV, Myocardium → Myo, Left Ventricle → LV. We adopt the identical data augmentation (*i.e.*, random rotation, random cropping, and horizontal flipping) for fair comparisons.

| | ACDC | | | | | | | | | | | |
| | 1 Labeled (1%) | | | | 3 Labeled (5%) | | | | 7 Labeled (10%) | | | |
| Method | Average | RV | Myo | LV | Average | RV | Myo | LV | Average | RV | Myo | LV |
|---|---|---|---|---|---|---|---|---|---|---|---|---|
| UNet-F [30] | 91.5/0.996 | 90.5/0.606 | 88.8/0.941 | 94.4/1.44 | 91.5/0.996 | 90.5/0.606 | 88.8/0.941 | 94.4/1.44 | 91.5/0.996 | 90.5/0.606 | 88.8/0.941 | 94.4/1.44 |
| UNet-L | 40.3/22.7 | 29.0/25.4 | 43.6/15.3 | 48.2/27.5 | 51.7/13.1 | 36.9/30.1 | 54.9/4.27 | 63.4/5.11 | 79.5/2.73 | 65.9/0.892 | 82.9/2.70 | 89.6/4.60 |
| EM [90] | 43.1/18.1 | 38.7/23.1 | 42.0/12.0 | 48.7/19.4 | 59.8/5.64 | 44.2/11.1 | 63.2/3.23 | 71.9/2.57 | 75.7/2.73 | 68.0/0.892 | 76.5/2.70 | 82.7/4.60 |
| CCT [91] | 48.6/19.2 | 38.7/28.0 | 49.2/14.8 | 57.9/17.0 | 59.1/10.1 | 44.6/19.8 | 63.2/6.04 | 69.4/4.32 | 75.9/3.60 | 67.2/2.90 | 77.5/3.32 | 82.9/4.59 |
| DAN [92] | 48.9/17.5 | 45.4/19.7 | 41.0/8.88 | 60.4/23.8 | 56.4/15.1 | 47.1/21.7 | 58.1/11.6 | 63.9/11.9 | 76.5/3.01 | 75.7/2.61 | 73.3/3.11 | 80.5/3.31 |
| URPC [57] | 43.0/21.1 | 38.6/22.8 | 41.7/14.4 | 48.6/26.0 | 58.9/8.14 | 50.1/12.6 | 60.8/4.10 | 65.8/7.71 | 83.1/1.68 | 77.0/0.742 | 82.2/0.505 | 90.1/3.79 |
| DTC [9] | 51.7/17.5 | 39.3/23.3 | 54.6/9.12 | 61.3/20.2 | 56.9/7.59 | 35.1/9.17 | 62.9/6.01 | 72.7/7.59 | 84.3/4.04 | 83.8/3.72 | 83.5/4.63 | 85.6/3.77 |
| DCT [93] | 49.7/16.4 | 42.4/20.4 | 48.8/10.6 | 57.9/18.2 | 58.5/10.8 | 41.2/21.4 | 63.9/5.01 | 70.5/6.05 | 78.1/2.64 | 70.7/1.75 | 77.7/2.90 | 85.8/3.26 |
| ICT [95] | 42.1/21.0 | 36.5/18.5 | 43.4/11.1 | 46.3/33.5 | 59.0/6.59 | 48.8/11.4 | 61.4/4.59 | 66.6/3.83 | 80.6/1.64 | 75.1/0.898 | 80.2/1.53 | 86.6/2.48 |
| MT [59] | 42.9/15.1 | 32.5/21.9 | 46.2/8.99 | 50.1/14.7 | 58.3/11.2 | 39.0/21.5 | 58.7/7.47 | 77.3/4.72 | 80.1/2.33 | 75.2/1.22 | 79.2/2.32 | 86.0/3.45 |
| UAMT [8] | 36.9/15.2 | 32.5/21.9 | 46.2/8.99 | 50.1/14.7 | 48.3/9.14 | 37.6/18.9 | 50.1/4.27 | 57.3/4.17 | 81.8/4.04 | 79.9/2.73 | 80.1/3.32 | 85.4/6.07 |
| SASSNet [94] | 42.6/24.8 | 29.8/34.7 | 45.4/13.3 | 52.5/26.6 | 57.8/6.36 | 47.9/11.7 | 59.7/4.51 | 65.8/2.87 | 84.7/1.83 | 81.8/0.769 | 82.9/1.73 | 89.4/2.99 |
| CPS [62] | 51.5/15.3 | 41.1/17.7 | 52.0/7.27 | 61.4/21.0 | 61.0/2.92 | 43.8/2.95 | 64.5/2.84 | 74.8/2.95 | 78.8/3.41 | 74.0/1.95 | 78.1/3.11 | 84.5/5.18 |
| GCL [50] | 59.7/14.3 | 49.5/25.3 | 60.9/6.28 | 68.8/11.5 | 70.6/2.24 | 56.5/1.99 | 70.7/1.67 | 84.8/3.05 | 87.0/0.751 | 86.9/0.584 | 81.8/0.820 | 92.5/0.849 |
| MC-Net [96] | 53.4/17.17 | 43.0/25.3 | 51.2/7.41 | 60.8/15.11 | 62.8/2.59 | 52.7/5.14 | 62.6/0.807 | 73.1/1.81 | 86.5/1.89 | 85.1/0.745 | 84.0/2.12 | 90.3/2.81 |
| SS-Net [53] | 63.4/2.94 | 64.7/3.32 | 57.0/1.81 | 68.4/3.70 | 65.8/2.28 | 57.5/3.91 | 65.7/2.02 | 74.2/0.896 | 86.8/1.40 | 85.4/1.19 | 84.3/1.44 | 90.6/1.57 |
| ACTION [16] | 81.0/3.45 | 76.9/3.09 | 78.4/2.07 | 87.5/5.17 | 86.6/1.20 | 85.2/0.734 | 84.7/0.909 | 89.8/1.97 | 87.2/1.47 | 86.1/0.976 | 85.7/1.11 | 89.7/2.33 |
| MONA [17] | 82.6/1.43 | 80.2/1.57 | 79.9/1.10 | 87.8/1.43 | 86.9/1.07 | 84.7/1.01 | 85.4/0.731 | 90.6/1.48 | 87.7/1.33 | 86.9/0.687 | 85.7/1.70 | 90.5/2.14 |
| • ARCO-SAG (ours) | 84.9/1.47 | 81.7/1.98 | 81.9/0.903 | **90.9**/1.52 | 87.1/0.848 | 85.6/0.414 | 85.1/0.930 | 90.6/**1.20** | 88.5/1.40 | 87.1/0.635 | 86.2/1.04 | **92.2**/2.53 |
| ○ ARCO-SG (ours) | **85.5**/**0.947** | **81.8**/**1.19** | **83.8**/**0.801** | **90.9**/**0.853** | **88.7**/**0.841** | **88.2**/0.618 | **85.9**/**0.673** | **91.9**/1.23 | **89.4**/0.776 | **90.2**/0.701 | **86.5**/0.787 | 91.6/**0.839** |

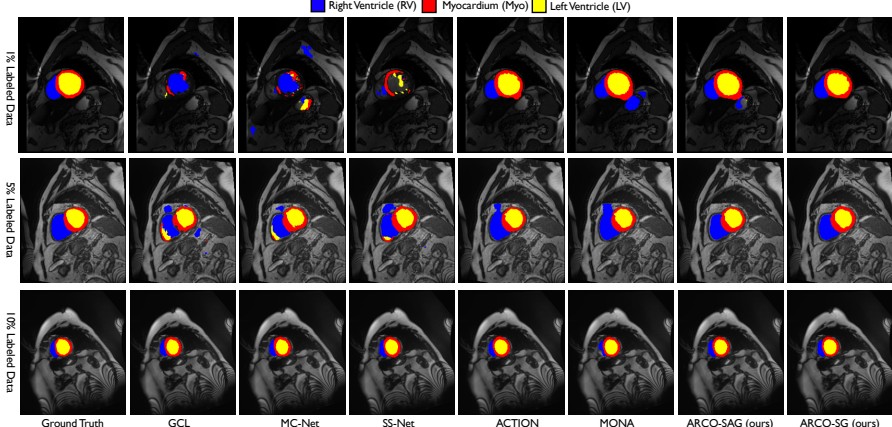

Figure 4: Visual results on ACDC with 1%, 5%, 10% label ratios. ARCO consistently produce more accurate predictions on anatomical regions and boundaries compared to all other SSL methods.

(*i.e.*, ACDC [81], LiTS [82], MMWHS [83]), two 3D benchmarks (*i.e.*, LA [84], in-house MP-MRI), and different label ratios (*i.e.*, 1%, 5%, 10%). As shown in Table 1, Table 5 (Appendix G), Table 6 (Appendix H), and Table 9 (Appendix I and J), we observe that our methods consistently outperform all the compared SSL-based methods by a considerable margin across all datasets and label ratios, which validates the superior performance of our proposed methods in both segmentation accuracy and label efficiency. For example, compared to the second-best MONA, our ARCO-SG under {1%, 5%, 10%} label ratios achieves {2.9%↑, 1.8%↑, 1.7%↑}, {3.3%↑, 1.8%↑, 1.8%↑}, {4.1%↑, 2.0%↑, 1.8%↑}, {0.3%↑, 0.3%↑, 0.5%↑}, {2.2%↑, 0.8%↑, 0.4%↑} in average Dice across ACDC, LiTS, and MMWHS, MP-MRI, and LA, respectively. Our ARCO-SAG achieves {84.9%, 87.1%, 88.5%}, {64.1%, 67.3%, 69.4%}, {86.1%, 88.6%, 89.3%}, {91.5%, 92.5%, 92.6%}, {73.2%, 86.9%, 89.1%} in averaged Dice across ACDC, LiTS, MMWHS, MP-MRI, and LA. These results indicate that our methods can generalize to different clinical scenarios and label ratios.

❷ **Across Label Ratios and Robustified Methods.** To further validate the label efficiency property of our ARCO, we evaluate our ARCO-SG and ARCO-SAG with limited labeled training data available (*e.g.*, 1% and 5%). As demonstrated in Table 1, Table 5 (Appendix G), Table 6 (Appendix H), and Table 9 (Appendix I and J), our models under 5% label ratios surpass all the compared SSL methods by a significant performance margin. For example, compared to MONA, we observe our methods to

Table 2: Ablation on component aspect: (1) tailness/$\mathcal{L}_{\text{contrast}}$; (2) diversity/$\mathcal{L}_{\text{nn}}$.

| Method | DSC[%]↑ | ASD[voxel]↓ |
|---|---|---|
| Vanilla | 49.3 | 7.11 |
| • ARCO-SAG (ours) | 84.9 | 1.47 |
| w/o tailness-SAG | 60.9 | 4.11 |
| w/o diversity | 78.6 | 1.68 |
| ○ ARCO-SG (ours) | 85.5 | 0.947 |
| w/o tailness-SG | 60.9 | 4.11 |
| w/o diversity | 79.3 | 1.26 |
| ARCO-NS | 82.6 | 1.43 |
| w/o tailness-NS | 60.9 | 4.11 |
| w/o diversity | 75.2 | 2.07 |

Table 3: Ablation on loss function: (1) unsupervised loss/$\mathcal{L}_{\text{unsup}}$; (2) global instance discrimination loss/$\mathcal{L}_{\text{inst}}^{\text{global}}$; and (3) local instance discrimination loss/$\mathcal{L}_{\text{inst}}^{\text{local}}$.

| Method | DSC[%]↑ | ASD[voxel]↓ |
|---|---|---|
| • ARCO-SAG (ours) | 84.9 | 1.47 |
| w/o $\mathcal{L}_{\text{unsup}}$ | 81.2 | 1.87 |
| w/o $\mathcal{L}_{\text{inst}}^{\text{global}}$ | 84.0 | 2.64 |
| w/o $\mathcal{L}_{\text{inst}}^{\text{local}}$ | 83.3 | 2.63 |
| ○ ARCO-SG (ours) | 85.5 | 0.947 |
| w/o $\mathcal{L}_{\text{unsup}}$ | 81.9 | 1.04 |
| w/o $\mathcal{L}_{\text{inst}}^{\text{global}}$ | 84.1 | 2.10 |
| w/o $\mathcal{L}_{\text{inst}}^{\text{local}}$ | 83.8 | 2.11 |

push the best segmentation accuracy higher by 0.3%∼2.0% in Dice on ACDC, LiTS, and MMWHS, MP-MRI, and LA, respectively. For example, the best segmentation accuracy on MMWHS rises from 87.3% to 89.3%. This suggests that our SSL-based approaches – without compromising the best achievable segmentation results – robustly improve performance using very limited labels, and further lead to a much-improved trade-off between SSL schemes and supervised learning schemes by avoiding a large amount of labeled data.

Similar to our results under 5% label ratio, our ARCO-SG and ARCO-SAG trained with 1% label ratio demonstrate sufficient performance boost compared to MONA by an especially significant margin, with up to 0.3%∼4.1% relative improvement in Dice. Taking the extremely limited label ratio (*i.e.*, 1%) as an indicator: (1) on 3D LA, ARCO-SG achieves 2.2% higher average Dice, and 6.64 lower average ASD than the second best MONA; (2) on LiTS, ARCO-SG achieves 3.3% higher average Dice, and 4.7 lower average ASD than the second best MONA; and (3) considering the more challenging clinical scenarios (*i.e.*, 7 anatomical classes), ARCO-SG achieves 5.1% higher average Dice, and 2.26 lower average ASD than the second-best MONA on MMWHS. It highlights the superior performance of ARCO is not only from improved label efficiency but also credits to the superior model robustness.

❸ **Qualitative Results.** We provide qualitative illustrations of ACDC, LiTS, MMWHS, LA, MP-MRI in Figure 4, Figure 6 (Appendix G), Figure 7 (Appendix H), Figure 8 (Appendix I), and Figure 9 (Appendix J), respectively. As shown in Figure 4, we observe that ARCO appears a significant advantage, where the edges and the boundaries of different anatomical regions are clearly more pronounced, such as RV and Myo regions. More interestingly, we found that in Figure 6, though all methods may confuse ambiguous tail-class samples such as small lesions, ARCO-SG and ARCO-SAG still produces consistently sharp and accurate object boundaries compared to the current approaches. We also observe similar results for ARCO on MMWHS in Figure 7 (Appendix H), where our approaches can regularize the segmentation results to be smooth and shape-consistent. Our findings suggest that ARCO improves model robustness mainly through distinguishing the minority tail-class samples.

## 4.2 Ablation Studies

In this subsection, we conduct various ablations to better understand our design choices. For all the ablation experiments the models are trained on ACDC with 1% labeled ratio.

**Importance of Loss Components.** We analyse several critical components of our method in the final performance and conduct comprehensive ablation studies on the ACDC dataset with a 1% label ratio to validate their necessity. **First**, at the heart of our method is the combination of three losses: $\mathcal{L}_{\text{contrast}}$ for *tailness*, and $\mathcal{L}_{\text{nn}}$ for *diversity* (See Section 3 for more details). We deactivate each component and then evaluate the resulting models, as shown in Table 2. As is shown, global contrastive loss $\mathcal{L}_{\text{contrast}}$ and nearest neighbor loss $\mathcal{L}_{\text{nn}}$ can boost performance by a large margin. Moreover, incorporating our methods (*i.e.*, SG and SAG) consistently achieve superior model robustness gains compared to naïve sampling (*i.e.*, NS), both of which suggests the importance of these components. **Second**, we compare the impact of different loss function (*e.g.*, unsupervised loss/$\mathcal{L}_{\text{unsup}}$, global instance discrimination loss/$\mathcal{L}_{\text{inst}}^{\text{global}}$, and local instance discrimination loss/$\mathcal{L}_{\text{inst}}^{\text{local}}$). As shown in Table 3, using each loss function consistently achieves significant performance gains, which demonstrates positive contribution of each component to performance gains.

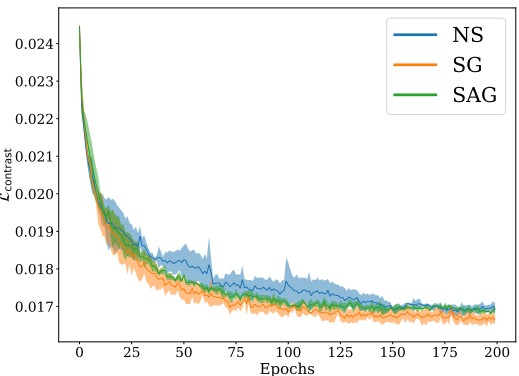

Figure 5: Visualization of training trajectories given by $\mathcal{L}_{\text{contrast}}$ vs. epochs on ACDC under 10% label ratio. The proposed ARCO is compared in terms of different sampling methods: Naïve Sampling (NS), Stratified Group (SG) Sampling, and Stratified-Antithetic Group (SAG) Sampling. The solid line and shaded area of each sampling method denote the mean and variance of test accuracies over 3 independent trials. Clearly, we observe SG sampling consistently outperforms the other sampling methods in convergence speed and training stability. SAG slightly outperforms NS.

**Importance of Augmentation Components.** We further investigate the impact of data augmentation on the ACDC dataset with a 1% label ratio. As is shown in Table 4 (in Appendix), employing each data augmentation strategy consistently results in notable performance improvements, underscoring the efficacy of these data augmentations. Of note, ARCO-SAG/ARCO-SG using all three augmentations and ARCO-SAG/ARCO-SG using no augmentation are considered as the upper bound and the lower bound for the performance comparison. These results show that each augmentation strategy systematically boosts performance by a large margin, which suggests improved robustness.

**Stability Analyses.** In Figure 5, we show the stability analysis results on ARCO over different sampling methods. As we can see, our SG and SAG sampling facilitates convergence during the training. More importantly, SG sampling has stable performance with small standard derivations, which aligns with our hypothesis that our proposed sampling method can be viewed as the form of variance regularization. Moreover, loss landscape visualization of different loss functions (Figure 3) reveals similar conclusions.

**Extra Study.** More investigations about (1) generalization across label ratios and frameworks in Appendix L; (2) final checkpoint loss landscapes in Appendix M; (3) ablation on different training settings are in Appendix N.

## 5 Conclusion and Discussion of Broader Impact

In this paper, we propose ARCO, a new semi-supervised contrastive learning framework for improved model robustness and label efficiency in medical image segmentation. Specifically, we propose two practical solutions via stratified group theory that correct for the variance introduced by the common sampling practice, and achieve significant performance benefits. Our theoretical findings indicate Stratified Group and Stratified-Antithetic Group Sampling provide practical means for improving variance reduction. It presents a curated and easily adaptable training toolkit for training deep networks that generalize well beyond training data in those long-tail clinical scenarios. Moreover, our sampling techniques can provide pragmatic solutions for enhancing variance reduction, thereby fostering their application in a wide array of real-world applications and sectors. These include but are not limited to 3D rendering, augmented reality, virtual reality, trajectory prediction, and autonomous driving. We hope this study could be a stepping stone towards by quantifying the limitation of current self-supervision objectives for accomplishing such challenging safety-critical tasks.

**Broader Impact.** Defending machine learning models against inevitable variance will have the great potential to build more reliable and trustworthy clinical AI. Our findings show that the stratified group theory can provide practical means for improving variance reduction, leading to realistic deployments in a large variety of real-world clinical applications. Besides, we should address the challenges of fairness or privacy in the medical image analysis domain as our future research direction.

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

# Appendix to
# Rethinking Semi-Supervised Medical Image Segmentation:
# A Variance-Reduction Perspective

**Sec. A** provides additional theoretical analysis.

**Sec. B** provides additional details on the training datasets.

**Sec. C** provides additional details on the implementation.

**Sec. D** provides details on 2D/3D methods in comparision.

**Sec. I** provides more experimental results on the 3D LA [84] dataset.

**Sec. K** provides more experimental results on three semantic segmentation benchmarks (Cityscapes [97], Pascal VOC 2012 [98], SUN RGB-D [99]).

## A  Theoretical Results Details

### A.1  Proof of Lemma 3.1

*Proof.* By definition of the estimate $\widehat{H}$, since the sampling from each group is independent, the variance can be written as

$$\mathrm{Var}[\widehat{H}_{\mathrm{SG}}] = \frac{1}{n^2} \sum_{m=1}^{M} \sum_{p \in \mathcal{D}_m} \sigma_m^2.$$

Since $|\mathcal{D}_m| = n_m$, it follows that

$$\mathrm{Var}[\widehat{H}_{\mathrm{SG}}] = \sum_{m=1}^{M} \frac{n_m^2}{n^2} \frac{\sigma_m^2}{n_m}.$$

Note that $\sigma_m^2$ is the variance of the $h(\mathbf{x}; p)$ where $p \sim \mathcal{P}_m$ uniformly.

**Analysis of SAG.** We now consider SAG. Recall that, compared to SG, SAG first samples $n_m/2$ pixels from $\mathcal{P}_m$[6] from each $m = 1, \cdots, M$ in the same way as SG. SAG then deterministically picks the rest $n_m/2$ pixels to be the reflection pixels of the first half (see Figure 2(3)). Let's choose arbitrary group $m$ and denote by $p$ and $p'$ the sampled pixel and its reflection from that group. Note the variance of $p$ satisfies $\mathrm{Var}_{\mathrm{SAG}}[h(\mathbf{x}; p)] = \sigma_m^2$ since $p$ is sampled in a same way as in the case of SG. Observe that, by symmetry, the variance of $p$ and $p'$ should be the same $\mathrm{Var}_{\mathrm{SAG}}[h(\mathbf{x}; p)] = \mathrm{Var}_{\mathrm{SAG}}[h(\mathbf{x}; p)] = \sigma_m^2$. It follows that:

$$\begin{aligned}
&\mathrm{Var}_{\mathrm{SAG}}[h(\mathbf{x}; p) + h(\mathbf{x}; p')] \\
&= \mathrm{Var}_{\mathrm{SAG}}[h(\mathbf{x}; p)] + \mathrm{Var}_{\mathrm{SAG}}[h(\mathbf{x}; p')] + 2\mathrm{Cov}_{\mathrm{SAG}}[h(\mathbf{x}; p), h(\mathbf{x}; p')] \\
&\leq 4\sigma_m^2,
\end{aligned}$$

---

[6]For simplicity, we ignore the rounding issue.

where the second step holds because the correlation between $h(\mathbf{x}; p)$ and $h(\mathbf{x}; p')$ is at most 1. It follows that:

$$
\begin{aligned}
\mathrm{Var}[\widehat{H}_{\mathrm{SAG}}] &= \sum_{m=1}^{M} \frac{|\mathcal{P}_m|^2}{|\mathcal{P}|^2} \frac{1}{|\mathcal{D}_m|^2} \mathrm{Var}\left[\sum_{p \in \mathcal{D}_m} h(\mathbf{x}; p)\right] \\
&= \sum_{m=1}^{M} \frac{n_m^2}{n^2} \frac{1}{n_m^2} \sum_{p, p'} \mathrm{Var}[h(\mathbf{x}; p) + h(\mathbf{x}; p')] \\
&\leq \sum_{m=1}^{M} \frac{n_m^2}{n^2} \frac{1}{n_m^2} \frac{n_m}{2} 4\sigma_m^2 \\
&= \sum_{m=1}^{M} \frac{2n_m}{n^2} \sigma_m^2 \\
&= 2\,\mathrm{Var}[\widehat{H}_{\mathrm{SG}}].
\end{aligned}
$$

$\square$

## A.2   Proof of Theorem 3.2

*Proof of Theorem 3.2.* Denote $w_m = n_m/n$, where $n_m = |\mathcal{D}_m|$. The unbiasedness is straightforward: by proportional sampling, the probability that an arbitrary pixel in group $\mathcal{P}_m$ being chosen is equal to $|\mathcal{D}_m|/|\mathcal{P}_m|$. Since $|\mathcal{D}_m| = |\mathcal{P}_m|$, the probability of being chosen is equal across all pixels, and hence an arbitrary pixel $p \in \mathcal{D} = \cup_m \mathcal{D}_m$ is equally likely to be any pixel in the population $\mathcal{P}$. As a result, we have

$$
\begin{aligned}
\mathbb{E}[\widehat{H}_{\mathrm{SG}}(\mathbf{x}; \mathcal{D})] &= \mathbb{E}\left[\sum_{p \in \mathcal{D}} h(\mathbf{x}; p)/|\mathcal{D}|\right] = \mathbb{E}_p[h(\mathbf{x}; p)] \\
&= H(\mathbf{x}),
\end{aligned}
$$

where the first equality is by the construction of $\widehat{H}_{\mathrm{SG}}(\mathbf{x}; \mathcal{D})$, the second equality is by symmetry of $p \in \mathcal{D}$, and the last inequality is by the above conclusion that $p$ is equally possible to be any pixel in $\mathcal{P}$. This finishes the proof that $\widehat{H}_{\mathrm{SG}}$ is unbiased.

The variance of $\widehat{H}_{\mathrm{SG}}$ is equal to

$$
\mathrm{Var}[\widehat{H}_{\mathrm{SG}}] = \sum_{m=1}^{M} \frac{n_m^2}{n^2} \frac{\sigma_m^2}{n_m},
$$

where $\sigma_m^2$ is the variance of the $f(\mathbf{x}; p)$ when $p \sim \mathcal{D}_m$ uniformly. Under the case of proportional sampling, i.e., $n_m \propto |\mathcal{D}_m|$ for all $m = 1, \cdots, M$ (for simplicity we assume such $n_m$'s are all integers), the variance becomes

$$
\mathrm{Var}[\widehat{H}_{\mathrm{SG}}] = \sum_{m=1}^{M} \frac{(w_m \cdot n)^2}{n^2} \frac{\sigma_m^2}{w_m \cdot n} = \sum_{m=1}^{M} w_m \frac{\sigma_m^2}{n}.
$$

We now consider NS (i.e. naïve sampling). By definition, the random sampling samples pixels uniformly from the set $\mathcal{P}$ of all pixels. Since $n$ is the total number of pixels sampled, the variance of the $\widehat{H}_{\mathrm{NS}}$ is given as $\mathrm{Var}[\widehat{H}_{NS}] = \sigma^2/n$ where $\sigma^2$ is the sampling variance of $h(\mathbf{x}; p)$ given $p \sim \mathcal{P}$ uniformly. To determine $\sigma^2$, we apply a variance decomposition trick via conditioning. Specifically, the sampling from NS is a two-step process: (1) sample the group index $m$ from $[M]$, (2) sample the pixel uniformly from $\mathcal{P}_m$. Applying the law of total variance, we have

$$
\begin{aligned}
&\mathrm{Var}_{p \overset{\mathrm{unif.}}{\sim} \mathcal{P}}[h(\mathbf{x}; p)] \\
&= \mathbb{E}[\mathrm{Var}[h(\mathbf{x}; p)]|p \in \mathcal{P}_m] + \mathrm{Var}[\mathbb{E}[h(\mathbf{x}; p)|p \in \mathcal{P}_m]] \\
&= \sum_{m=1}^{M} w_m \sigma_m^2 + \sum_{m=1}^{M} \left(\mathbb{E}_{p \overset{\mathrm{unif.}}{\sim} \mathcal{P}_m}[h(\mathbf{x}; p)] - \mathbb{E}_{p \overset{\mathrm{unif.}}{\sim} \mathcal{P}}[h(\mathbf{x}; p)]\right)^2.
\end{aligned}
$$

As a result, we conclude that, for any image $\mathbf{x}$, the sampling variance of SG estimate and NS estimate satisfies

$$\mathrm{Var}[\widehat{H}_{\mathrm{NS}}] = \mathrm{Var}[\widehat{H}_{\mathrm{SG}}]$$
$$+ \frac{1}{n} \sum_{m=1}^{M} \left( \mathbb{E}_{p \overset{\mathrm{unif.}}{\sim} \mathcal{P}_m}[h(\mathbf{x}; p)] - \mathbb{E}_{p \overset{\mathrm{unif.}}{\sim} \mathcal{P}}[h(\mathbf{x}; p)] \right)^2,$$

which finishes the proof. $\qquad\square$

### A.3 Further Details on Convergence

We first make the following assumptions about the loss function and the gradient estimate in our learning problem.

**Assumption A.1** (Smoothness). *The objective function $\mathcal{L}(\cdot)$ is $L$-smooth, i.e., $\mathcal{L}$ is differentiable and $\|\nabla \mathcal{L}(\theta) - \nabla \mathcal{L}(\theta')\|_2 \leq L(\|\theta - \theta'\|_2)$ for all $\theta$, $\theta'$ in the domain of $\nabla \mathcal{L}$.*

**Assumption A.2** (Bounded variance). *There exists some $\sigma_g^2 > 0$ such that for all $\theta$ in the domain,*

$$\mathbb{E}\left[ \|g(\theta) - \nabla \mathcal{L}(\theta)\|^2 \right] \leq \sigma_g^2.$$

These assumptions are standard in optimization theory and in various settings [87, 88, 89]. The first assumption allows us to characterize the loss landscape of our learning problem, and the second assumption ensures the gradient estimate does not deviate largely from the truth, which is essential for the training to converge. With these two assumptions, it is guaranteed that

$$\frac{1}{T} \sum_{t=1}^{T} \mathbb{E}\left[ \|\nabla \mathcal{L}(\theta)\|_2^2 \right] \leq C \left( \frac{1}{T} + \frac{\sigma_g}{\sqrt{T}} \right).$$

**Proposition 1.** *Suppose the step sizes in SGD are taken to be $\min\left\{ 1/L, \alpha/(\sigma_g \sqrt{T}) \right\}$ where $T$ is the number of steps. Then under Assumption A.1 and A.2, there exists some constant $C$ depending on $\mathcal{L}$, $\alpha$, and $L$, such that*

$$\frac{1}{T} \sum_{t=1}^{T} \mathbb{E}\left[ \|\nabla \mathcal{L}(\theta_t)\|_2^2 \right] \leq C \left( \frac{1}{T} + \frac{\sigma_g}{\sqrt{T}} \right).$$

Proposition 1 shows that, in expectation, the convergence rate of the average gradient norm consists of a fast rate $1/T$ and a slow rate $\sigma_g/\sqrt{T}$, with the slow rate being the dominant term. Importantly, the slow rate depends on the variance of the gradient estimate. This suggests that a variance reduced gradient estimate allows SGD to reach an approximate local minimum with less iterations, leading to faster convergence of the training loss. This intuition is corroborated by Fig. 5, where we directly visualize the training trajectory of contrastive loss versus epoch for three sampling methods. We observe that SG features a faster loss decay and a narrower error bar, showing that it outperforms other methods in both the convergence speed and the stability. Furthermore, Proposition 1 indicates that our proposed sampling techniques are universal, since it applies to all scenarios as long as the mild assumptions A.1 and A.2 are satisfied. In other words, when it comes to scenarios that involve pixel/voxel-level sampling on 2D/3D images, utilizing SG/SAG instead of naive sampling can lead to enhanced stability and decreased variance.

## B  Datasets

Our experiments are conducted on five 2D/3D representative datasets in semi-supervised medical image segmentation literature, including 2D benchmarks (*i.e.*, ACDC [81], LiTS [82], and MMWHS [83]) and 3D benchmarks (*i.e.*, LA [84] and in-house MP-MRI).

- **The ACDC dataset** was collected from MICCAI 2017 ACDC challenge [81], consisting of 200 3D cardiac cine MRI scans with 3 classes – left ventricle (LV), myocardium (Myo), and right ventricle (RV). Following [53, 9, 10], we utilize 120, 40, and 40 scans for training,

validation, and testing [7]. Note that 1%, 5%, and 10% label ratios are the ratio of patients. The splitting details are in the supplementary material. For pre-processing, we normalize the intensity of each 3D scan into $[0, 1]$ by using min-max normalization, and re-sample images and segmentation maps to $256 \times 256$ pixels.

- **The LiTS dataset** was collected from MICCAI 2017 Liver Tumor Segmentation Challenge [82], consisting of 131 contrast-enhanced 3D abdominal CT volumes with 2 classes – liver and tumor. Note that 1%, 5%, and 10% label ratios are the ratio of patients. We utilize 100 and 31 scans for training and testing, with random order. The splitting details are in the supplementary material. For pre-processing, we follow the setting in [6] by truncating the intensity of each 3D scan into $[-200, 250]$ HU for removing irrelevant and redundant features, normalizing each 3D scan into $[0, 1]$, and re-sample images and segmentation maps to $256 \times 256$ pixels.

- **The MMWHS dataset** was collected from MICCAI 2017 challenge [83], consisting of 20 3D cardiac MRI scans with 7 classes – left ventricle (LV), left atrium (LA), right ventricle (RV), right atrium (RA), myocardium (Myo), ascending aorta (AAo), and pulmonary artery (PA). Note that 1%, 5%, and 10% label ratios are based on the ratio of patients. Following [17], we utilize 15 and 5 scans for training and testing. The splitting details are in the supplementary material. For pre-processing, we normalize the intensity of each 3D scan into $[0, 1]$ by using min-max normalization, and re-sample images and segmentation maps to $256 \times 256$ pixels.

- **The LA dataset** [84] was a representative 3D benchmark, consisting of 100 gadolinium-enhanced MRI scans with one class – left atrium (LA), with an isotropic resolution of $0.625 \times 0.625 \times 0.625 \text{mm}^3$. Note that 1%, 5%, and 10% label ratios are the ratio of patients. The fixed split (*i.e.*, 5%, and 10% ) [8, 53] uses 80 and 20 scans for training and testing [8], and 1% label ratio is randomly split. The splitting details are in the supplementary material. For pre-processing, we crop all the scans at the heart region, normalize the intensities of each 3D scan into $[0, 1]$, and randomly crop all the training sub-volumes into $112 \times 112 \times 80 \text{mm}^3$.

- **The Multi-phasic MRI (MP-MRI) dataset** was an in-house 3D dataset, consisting of 160 multi-phasic MRI scans with one class – liver, each of which includes T1 weighted DCE-MRI images at three-time points (*i.e.*, pre-contrast, arterial phase, and venous phases). Three images are mutually registered to the arterial phase images, with an isotropic voxel size of $1.00 \times 1.00 \times 1.00 \text{mm}^3$. The dataset is randomly divided into 100 scans for training, 40 for validation, and 20 for testing. Note that 1%, 5%, and 10% label ratios are the ratio of patients. The splitting details are in the supplementary material. For pre-processing, we normalize the intensity of each 3D scan into $[0, 1]$ by using min-max normalization, and re-sample images and segmentation maps to $256 \times 256$ pixels.

## C  Implementation Details

In our experiments, all of our evaluated methods have been trained using similar settings for simplicity in reproducing our results. All experiments are conducted with PyTorch [101] on an NVIDIA RTX 3090 Ti. We adopt an SGD optimizer with momentum 0.9 and weight decay $10^{-4}$. The initial learning rate is set to 0.01. We use the 2D `UNet` [30] or 3D `VNet` [78] backbones as the segmentation network under different labeled ratio settings (*i.e.*, 1%, 5%, 10% labeled ratios). Following [17], we adopt the feature pyramid network (`FPN`) [102] architecture as the representation head $\psi_r$ with a separate 512-dimension output layer. The momentum hyperparameter is set to 0.99.

For pre-training, the networks are trained for 100 epochs with a batch size of 6. We apply the *mined* views with $d = 5$, following [17]. As for fine-tuning, the networks are trained for 200 epochs with a batch size of 8. The learning rate decays by a factor of 10 every 2500 iterations during the training. We apply the temperature with $\tau_t = 0.01$, $\tau_s = 0.1$, and $\tau = 0.5$, respectively. The size of the memory bank is set to 36. For the CL training, we use the implementation from [17] and leave all parameters on their default settings, *e.g.*, we apply the hyperparameters with $\lambda_1 = 0.01$, $\lambda_2 = 1.0$, and $\lambda_3 = 1.0$. For other hyper-parameters in all the evaluated methods, we adopt the suggested settings in the original papers because they are not of direct interest to us.

---

[7] https://github.com/HiLab-git/SSL4MIS/tree/master/data/ACDC
[8] https://github.com/ycwu1997/SS-Net/tree/main/data/LA

Table 4: Ablation on data augmentation strategies: (1) random rotation; (2) random cropping; and (3) horizontal flipping, compared to our methods with two settings (*i.e.*, w/ no augmentation and w/ all three augmentation). Note that we use the identical data augmentation strategies (*i.e.*, random rotation, random cropping, and horizontal flipping), as [30, 90, 91, 92, 93, 8, 94, 9, 57, 95, 62, 50, 59, 96, 53, 16, 17] for fair comparisons.

| Method | DSC[%]↑ | ASD[voxel]↓ |
|---|---|---|
| • ARCO-SAG (ours) | 84.9 | 1.47 |
| w/o random rotation | 84.1 | 2.87 |
| w/o random cropping | 84.4 | 3.47 |
| w/o horizontal flipping | 73.7 | 6.70 |
| w/ no augmentation | 70.8 | 9.83 |
| ○ ARCO-SG (ours) | 85.5 | 0.947 |
| w/o random rotation | 83.8 | 3.27 |
| w/o random cropping | 84.6 | 1.62 |
| w/o horizontal flipping | 78.8 | 4.15 |
| w/ no augmentation | 76.2 | 7.74 |

For 2D medical segmentation, we follow the same data augmentations [9, 53] to the *teacher*'s input and the *student*'s input, respectively. The augmentations include random rotation, random cropping, and horizontal flipping. For 3D medical segmentation, we follow the same data augmentations [8, 9, 96, 53] to the *teacher*'s input and the *student*'s input, respectively. The augmentations include random rotation and random flipping. We evaluate our methods on 3D segmentation results with two classical metrics: (1) Dice coefficient (DSC) and (2) Average Symmetric Surface Distance (ASD). Note that, for all the evaluated methods, we make no additional modifications during the training process for fair evaluations. We run all our experiments in the same environments with fixed random seeds (Hardware: Single NVIDIA GeForce RTX 3090 GPU; Software: PyTorch 1.10.2+cu113, and Python 3.8.11).

# D    2D/3D Methods in Comparison

**2D Medical Segmentation**: For experiments, we use 2D UNet [30] as backbone, and compare ARCO with previous state-of-the-art medical segmentation methods: (1) *Baseline* (*i.e.*, UNet-F/UNet-L [30]) using both fully-supervised and limited-supervised settings; and (2) *SSL-based*: EM [90], CCT [91], DAN [92], URPC [57], DTC [9], DCT [93], ICT [95], MT [59], UAMT [8], SASSNet [94], CPS [62], GCL [50], MC-Net [96], SS-Net [53], ACTION [16], and MONA [17]. Note that among all the above evaluated methods, several methods use a contrastive objective, including GCL [50], SS-Net [53], ACTION [16], and MONA [17].

**3D Medical Segmentation**: For experiments, we use 3D VNet [78] as backbone, and compare ARCO with previous state-of-the-art medical segmentation methods: (1) *Baseline* (*i.e.*, VNet-F/VNet-L [78]) using both fully-supervised and limited-supervised settings; and (2) *SSL-based*: EM [90], CCT [91], DAN [92], URPC [57], DTC [9], DCT [93], ICT [95], MT [59], UAMT [8], SASSNet [94], CPS [62], GCL [50], MC-Net [96], SS-Net [53], ACTION [16], and MONA [17]. Note that among all the above evaluated methods, several methods use a contrastive objective, including GCL [50], SS-Net [53], ACTION [16], and MONA [17].

# E    Framework Overview

In the following, we provide a concise overview of ARCO, consisting of two training phases: (1) relational semi-supervised pre-training, and (2) anatomical contrastive fine-tuning. Note that, in this paper, sampling strategies in pixel-level CL framework are of direct interest, so we use a simplification of MONA [17] without using any additional complicated augmentation strategies.

**(1) Relational Semi-Supervised Pretraining.** Given an unlabeled sample, ARCO has two ways to define *augmented* and *mined* views: (1) ARCO augments the samples to be $\mathbf{x}^1$ and $\mathbf{x}^2$ as *augmented* views, with two separate data augmentation operators; and (2) ARCO randomly samples $d$ *mined* views (*i.e.*, $\mathbf{x}^3$) from the unlabeled dataset with additional augmentation. The pairs $[\mathbf{x}^1, \mathbf{x}^2]$ are then processed by $[F_s, F_t]$, and in a similar way $\mathbf{x}^3$ is processed by $F_t$ (See Figure 1(a) in main paper), outputting three global features $[\mathbf{h}^1, \mathbf{h}^2, \mathbf{h}^3]$ after $E$ and their local features $[\mathbf{f}^1, \mathbf{f}^2, \mathbf{f}^3]$ after

Table 5: Quantitative comparisons (DSC[%] ↑ / ASD[voxel] ↓) across the three labeled ratio settings (1%, 5%, 10%) on the LiTS benchmark. All experiments are conducted as [30, 90, 91, 92, 93, 8, 94, 9, 57, 95, 62, 50, 59, 96, 53, 16, 17] in the identical setting for fair comparisons. Best and second-best results are coloured **blue** and red, respectively. UNet-F (fully-supervided) and UNet-L (semi-supervided) are considered as the upper bound and the lower bound for the performance comparison. Please refer to the text for discussion. We adopt the identical data augmentation (*i.e.*, random rotation, random cropping, and horizontal flipping) for fair comparisons.

| | LiTS | | | | | | | | |
|---|---|---|---|---|---|---|---|---|---|
| | 1% Labeled | | | 5% Labeled | | | 10% Labeled | | |
| Method | Average | Liver | Lesion | Average | Liver | Lesion | Average | Liver | Lesion |
| UNet-F [30] | 68.2/ 16.9 | 90.6/8.14 | 45.8/25.6 | 68.2/16.9 | 90.6/8.14 | 45.8/25.6 | 68.2/16.9 | 90.6/8.14 | 45.8/25.6 |
| UNet-L | 50.8/30.4 | 76.5/15.2 | 25.0/45.6 | 63.4/30.4 | 90.5/9.84 | 36.4/50.9 | 64.6/28.3 | 88.4/18.6 | 40.9/37.9 |
| EM [90] | 56.6/38.4 | 86.4/26.3 | 26.9/50.5 | 61.2/33.3 | 87.6/9.47 | 34.7/57.1 | 62.9/38.5 | 87.4/21.3 | 38.3/55.7 |
| CCT [91] | 52.4/52.3 | 79.1/42.0 | 25.7/62.6 | 61.4/26.1 | 84.6/12.3 | 38.2/39.9 | 63.4/23.0 | 88.8/7.64 | 38.0/38.5 |
| DAN [92] | 57.1/28.3 | 84.5/19.2 | 29.6/37.4 | 62.3/25.8 | 88.6/9.64 | 36.1/42.1 | 63.2/30.7 | 87.3/15.4 | 39.1/46.1 |
| URPC [57] | 55.5/34.6 | 83.0/27.3 | 28.0/42.0 | 60.9/21.4 | 85.5/8.11 | 36.3/34.6 | 62.5/24.8 | 84.9/12.9 | 40.1/36.7 |
| DTC [9] | 39.3/37.5 | 68.1/10.7 | 10.5/64.3 | 59.2/18.6 | 85.0/7.54 | 33.3/29.7 | 62.5/24.8 | 84.9/12.9 | 40.1/36.7 |
| DCT [93] | 57.7/38.5 | 87.0/22.4 | 28.3/54.6 | 60.8/34.4 | 89.2/12.6 | 32.5/56.2 | 61.9/31.7 | 86.2/19.3 | 37.5/44.1 |
| ICT [95] | 58.3/32.2 | 86.5/22.8 | 30.1/41.5 | 60.1/39.1 | 86.8/12.6 | 33.3/65.6 | 62.5/32.4 | 88.1/16.7 | 36.9/48.2 |
| MT [59] | 54.7/24.7 | 83.1/10.2 | 26.3/39.1 | 60.9/23.7 | 87.5/**6.34** | 34.4/41.1 | 62.3/23.7 | 88.5/9.32 | 36.1/38.1 |
| UAMT [8] | 55.5/34.6 | 83.0/27.3 | 28.0/42.0 | 61.5/24.7 | 84.5/10.6 | 38.6/38.8 | 62.9/23.6 | 87.4/7.78 | 38.4/39.6 |
| SASSNet [94] | 39.6/42.7 | 69.0/14.7 | 10.3/7.06 | 60.4/25.3 | 86.1/11.6 | 34.7/39.0 | 62.4/21.1 | 86.4/8.31 | 38.3/33.9 |
| CPS [62] | 57.7/39.6 | 87.0/22.4 | 28.3/54.6 | 59.5/26.3 | 84.0/9.01 | 34.9/43.5 | 62.1/30.8 | 88.6/18.3 | 35.6/43.3 |
| GCL [50] | 59.3/29.5 | 88.6/14.2 | 30.0/44.9 | 63.3/20.1 | 90.7/9.46 | 35.9/30.8 | 65.0/37.2 | 91.3/10.0 | 38.7/64.3 |
| MC-Net [96] | 60.9/32.1 | 87.1/17.8 | 34.8/46.5 | 61.6/19.8 | 86.3/8.21 | 36.9/31.4 | 63.4/29.9 | 89.0/13.1 | 38.0/46.8 |
| SS-Net [53] | 55.0/35.9 | 89.6/19.8 | 20.5/51.9 | 59.1/24.6 | 87.5/11.2 | 30.6/38.1 | 63.4/19.8 | 91.1/7.33 | 35.8/32.2 |
| ACTION [16] | 61.0/24.5 | 89.8/16.9 | 32.2/32.3 | 66.3/23.6 | 93.0/6.41 | 39.5/40.8 | 67.2/20.4 | 92.8/5.08 | 41.6/35.8 |
| MONA [17] | 62.2/24.9 | 91.3/13.9 | 34.0/35.9 | 66.6/16.6 | 93.1/7.74 | 40.1/25.4 | 68.3/18.0 | 93.4/8.88 | 43.3/27.0 |
| ● ARCO-SAG (ours) | 64.1/20.6 | 91.5/7.63 | 36.8/**33.5** | 67.3/13.2 | 93.0/7.10 | 41.7/19.3 | 69.4/14.9 | 93.3/7.60 | 45.5/22.2 |
| ○ ARCO-SG (ours) | **65.5**/**20.2** | **92.5**/**5.59** | **38.4**/34.7 | **68.4**/**11.3** | **93.7**/6.63 | **43.0**/**16.0** | **70.1**/**13.5** | **94.1**/**5.01** | **46.1**/**22.1** |

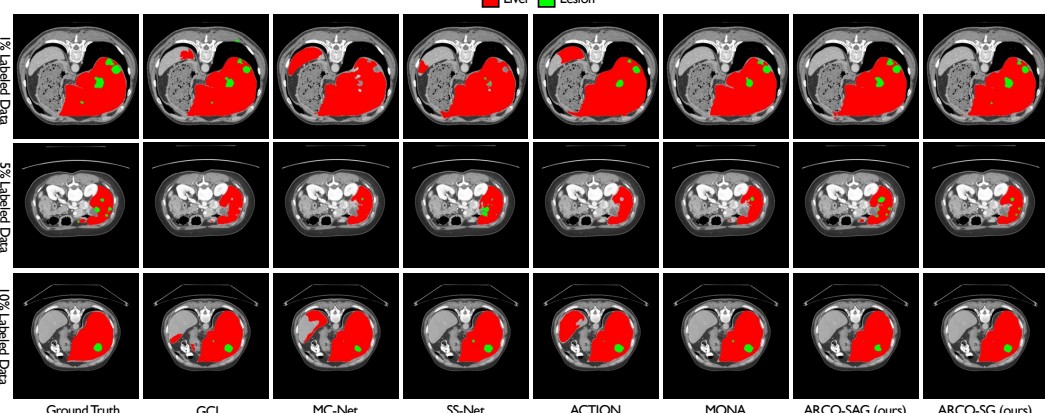

Figure 6: Visual results on LiTS with 1%, 5%, 10% label ratios. As is shown, ARCO-SG and ARCO-SAG consistently produce more accurate predictions on anatomical regions and boundaries compared to all other SSL methods.

$D$. These features are fed to the two-layer non-linear projectors for outputting global and local embeddings $\mathbf{v}_g$ and $\mathbf{v}_l$.

To alleviate the *collapse* issues [26, 27, 16], we make the architecture asymmetric between the *student* and *teacher* pipeline by further feeding both the global and local embeddings $\mathbf{v}$ with respect to the *student* branch into the non-linear predictor, producing $\mathbf{w}$ in both global and local manners [9]. After passing through the nonlinear projectors and predictor, the relational similarities between *augmented* and *mined* embeddings are computed using the softmax transform, which can be formulated as:

$$\mathbf{u}_s = \log \frac{\exp\left(\text{sim}\left(\mathbf{w}^1, \mathbf{v}^3\right)/\tau_s\right)}{\sum_{n=1}^{N} \exp\left(\text{sim}\left(\mathbf{w}^1, \mathbf{v}_n^3\right)/\tau_s\right)}, \quad \mathbf{u}_t = \log \frac{\exp\left(\text{sim}\left(\mathbf{w}^2, \mathbf{v}^3\right)/\tau_t\right)}{\sum_{n=1}^{N} \exp\left(\text{sim}\left(\mathbf{w}^2, \mathbf{v}_n^3\right)/\tau_t\right)},$$ where $\tau_s$ and $\tau_t$ are different temperature parameters. The unsupervised instance discrimination loss (*i.e.*, Kullback-Leibler divergence $\mathcal{KL}$) can be defined as:

$$\mathcal{L}_{\text{inst}} = \mathcal{KL}(\mathbf{u}_s \| \mathbf{u}_t). \tag{E.1}$$

---

[9]We omit details of local instance discrimination setting for simplicity in following contexts.

The parameters of the *teacher* model are the exponential moving average (EMA) of the *student* model parameters that are updated by the stochastic gradient descent. For pretraining, the entire loss consists of the global and local instance discrimination loss, and supervised segmentation loss $\mathcal{L}_{\text{sup}}$ (*i.e.*, equal combination of Dice loss and cross-entropy loss), *i.e.*, $\mathcal{L} = \mathcal{L}_{\text{inst}}^{\text{global}} + \mathcal{L}_{\text{inst}}^{\text{local}} + \mathcal{L}_{\text{sup}}$.

**(2) Anatomical Contrastive Finetuning (ACF).** We use the pre-trained network weights as the initialization for subsequent fine-tuning (See Figure 1(b) in Main Paper). For semi-supervised training, we follow two principles described in [17] [10]: (1) *tailness*: giving more importance to tail class hard pixels; and (2) *diversity*: ensuring anatomical diversity in the set of different sampled images.

Following the abovementioned principles, we employ a two-step routine: (1) *Tailness*: we first perform anatomical contrastive formulation. Specifically, we additionally attach the representation head $\psi_r$[11], and generate a higher $n$-dimensional dense representation with the same spatial resolution as the input image. A pixel-level contrastive loss is designed to pull queries $\mathbf{r}_q \in \mathcal{R}$ to be similar to the positive keys $\mathbf{r}_k^+ \in \mathcal{R}$, and push apart the negative keys $\mathbf{r}_k^- \in \mathcal{R}$. The semi-supervised contrastive loss $\mathcal{L}_{\text{contrast}}$ is defined as:

$$\mathcal{L}_{\text{contrast}} = \sum_{c \in \mathcal{C}} \sum_{\mathbf{r}_q \sim \mathcal{R}_q^c} - \log \frac{\exp(\mathbf{r}_q \cdot \mathbf{r}_k^{c,+}/\tau)}{\exp(\mathbf{r}_q \cdot \mathbf{r}_k^{c,+}/\tau) + \sum_{\mathbf{r}_k^- \sim \mathcal{R}_k^c} \exp(\mathbf{r}_q \cdot \mathbf{r}_k^-/\tau)}, \quad \text{(E.2)}$$

where $\mathcal{C}$ is a set including all available classes in the current mini-batch, and $\tau$ is a temperature hyperparameter. We refer to $\mathcal{R}_q^c, \mathcal{R}_k^c, \mathbf{r}_k^{c,+}$ as a query set including all representations within this class $c$, a negative key set including all representations whose labels is not in class $c$, and the positive key which is the $c$-class mean representation, respectively. Consider $\mathcal{P}$ is a set including all pixel coordinates with the same resolution as $R$, these queries and keys are then defined as:

$$\mathcal{R}_q^c = \bigcup_{[i,j] \in \mathcal{A}} \mathbb{1}(\mathbf{y}_{[i,j]} = c) \, \mathbf{r}_{[i,j]}, \ \mathcal{R}_k^c = \bigcup_{[i,j] \in \mathcal{A}} \mathbb{1}(\mathbf{y}_{[i,j]} \neq c) \, \mathbf{r}_{[i,j]}, \ \mathbf{r}_k^{c,+} = \frac{1}{|\mathcal{R}_q^c|} \sum_{\mathbf{r}_q \in \mathcal{R}_q^c} \mathbf{r}_q. \quad \text{(E.3)}$$

(2) *Diversity*: we leverage the first-in-first-out (FIFO) memory bank [13] to search for $K$-nearest neighbors, and use the semi-supervised nearest neighbor loss $\mathcal{L}_{\text{nn}}$ in a way that maximizing cosine similarity, to exploit the inter-instance relationship.

For fine-tuning, the total loss includes contrastive loss $\mathcal{L}_{\text{contrast}}$, nearest neighbors loss $\mathcal{L}_{\text{nn}}$, unsupervised cross-entropy loss $\mathcal{L}_{\text{unsup}}$ and supervised segmentation loss $\mathcal{L}_{\text{sup}}$: $\mathcal{L}_{\text{sup}} + \lambda_1 \mathcal{L}_{\text{contrast}} + \lambda_2 \mathcal{L}_{\text{unsup}} + \lambda_3 \mathcal{L}_{\text{nn}}$. See Appendix N for the ablation of hyperparameters.

# F  Model Architecture

Figure 1 provides an overview over our approach. Our semi-supervised segmentation model $F$ takes an 2D/3D medical image $x$ as input and outputs the segmentation map and the representation map. We leverage MONA pipeline [17] including two stages: (1) relational semi-supervised pre-training: on labeled data, the *student* network is trained by the ground-truth labels with the supervised loss $\mathcal{L}_{\text{sup}}$; while on unlabeled data, the *student* network takes the *augmened* and *mined* embeddings from the EMA teacher for instance discrimination $\mathcal{L}_{\text{inst}}$ in the global and local manner, (2) our proposed anatomical contrastive reconstruction fine-tuning: on labeled data, the student network is trained by the ground-truth labels with the supervised loss $\mathcal{L}_{\text{sup}}$; while on unlabeled data, the student model takes the representation maps and pseudo labels from the EMA teacher to give more importance to tail class $\mathcal{L}_{\text{contrast}}$, exploit the inter-instance relationship $\mathcal{L}_{\text{nn}}$, and compute unsupervised loss $\mathcal{L}_{\text{unsup}}$. See Appendix M for details of the visualization loss landscapes.

# G  More Experiment Results on LiTS

Figure 6 and Table 5 show qualitative and quantitative results, where our `ARCO-SG` and `ARCO-SAG` provide better segmentation results than all other methods. This clearly demonstrates the superiority of our models.

---

[10]In this paper, sampling strategies in pixel-level contrastive learning frameworks are of direct interest, so we use a simplification of `MONA` [17] – without using any additional complicated augmentation strategies.

[11]The representation head is only applied during training, and is removed during inference

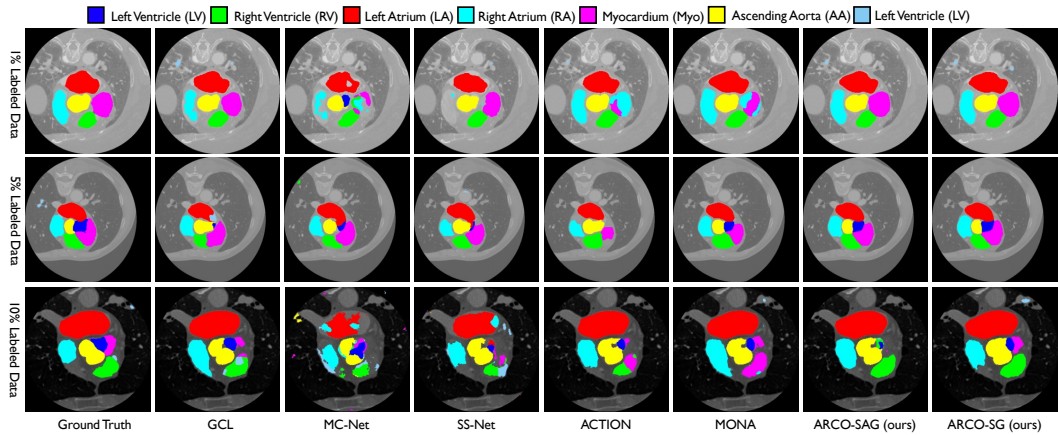

Figure 7: Visual results on MMWHS with 1%, 5%, 10% label ratios. As is shown, `ARCO-SG` and `ARCO-SAG` consistently produce more accurate predictions on anatomical regions and boundaries compared to all other SSL methods.

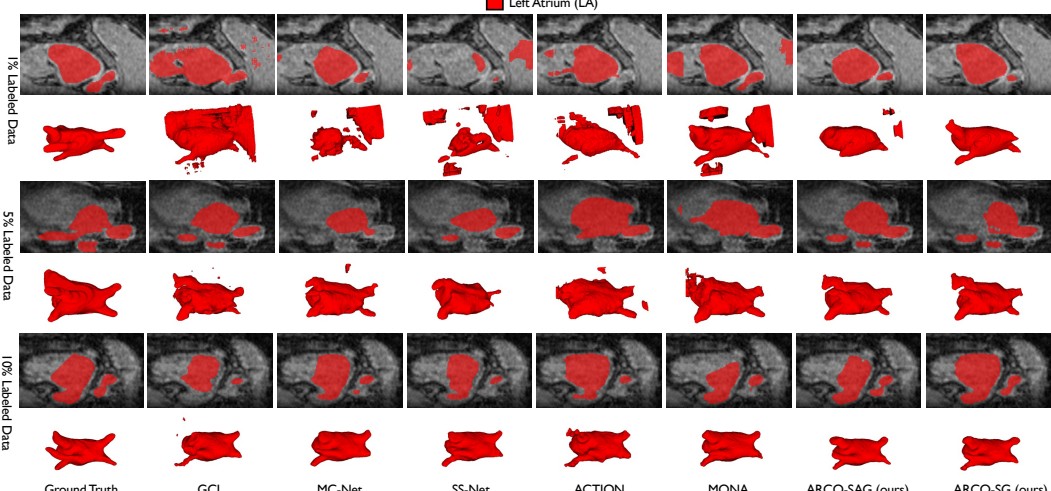

Figure 8: Visual results on LA with 1%, 5%, 10% label ratios. As is shown, `ARCO-SG` and `ARCO-SAG` consistently yield more accurate and sharper boundaries compared to all other SSL methods.

## H    More Experiment Results on MMWHS

We run the baselines and our methods on the third medical image segmentation dataset (*i.e.*, MMWHS [83]) under various label ratios (*i.e.*, 1%, 5%, 10%), reporting results in Table 6 and Figure 7. This clearly demonstrates the effectiveness of our models.

## I    More 3D Experiment Results on LA [84]

We run the baselines and our methods on the fourth 3D medical image segmentation dataset (*i.e.*, LA [84]) under various label ratios (*i.e.*, 1%, 5%, 10%), reporting results in Table 9 and Figure 8. This clearly demonstrates the robustness of our models.

## J    More 3D Experiment Results on MP-MRI

We run the baselines and our methods on the five 3D medical image segmentation dataset (*i.e.*, MP-MRI) under various label ratios (*i.e.*, 1%, 5%, 10%), reporting results in Table 9 and Figure 9. This clearly shows the superiority of our models.

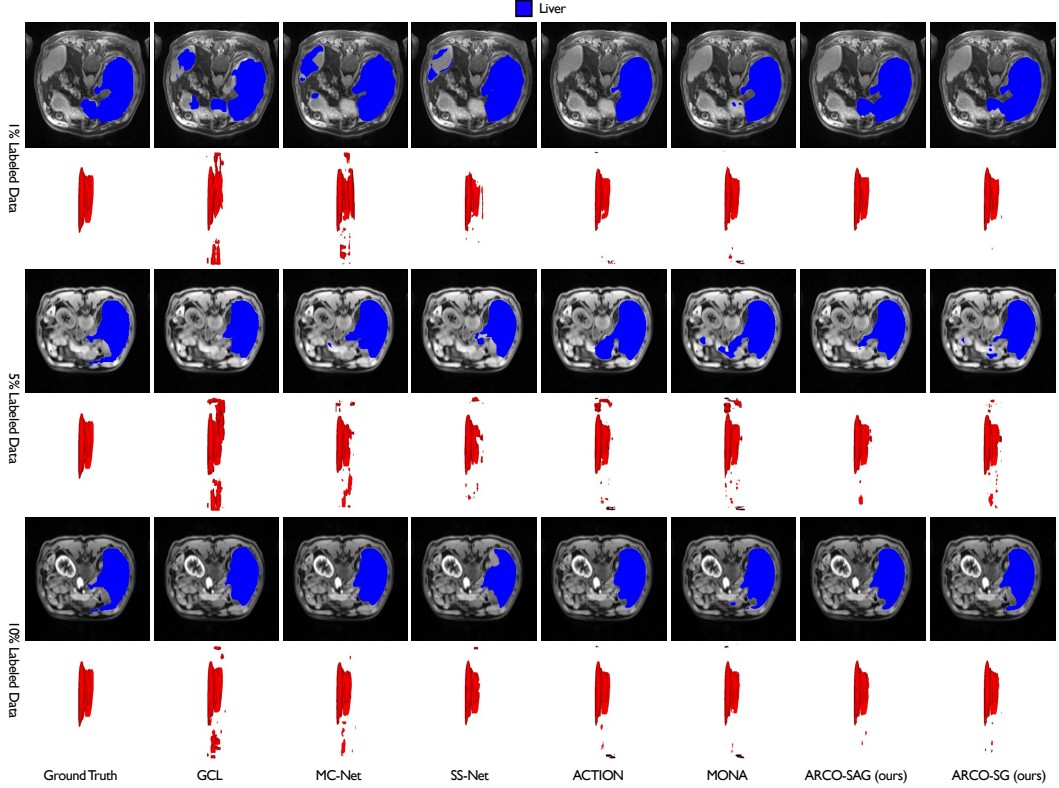

Figure 9: Visual results on MP-MRI with 1%, 5%, 10% label ratios. As is shown, `ARCO-SG` and `ARCO-SAG` consistently yield more accurate and sharper boundaries compared to all other SSL methods.

## K  More Experiment Results on Semantic Segmentation

To further validate the effectiveness, we experiment on three popular segmentation benchmarks (*i.e.*, Cityscapes [97], Pascal VOC 2012 [98], indoor scene segmentation dataset – SUN RGB-D [99]) in the semi-supervised full-label settings. We follow the identical setting [100] to sample labelled images to ensure that every class appears sufficiently in our three datasets, *i.e.*, CityScapes, Pascal VOC, and SUN RGB-D. Specifically, we aim to have the least frequent class appear in at least 5, 15, and 50 images in each dataset, respectively. In CityScapes, we have at least 12 semantic classes represented in our labeled images, while in Pascal VOC and SUN RGB-D we have at least 3 and 1 semantic classes, respectively. We compare our `ARCO-SG` and `ARCO-SAG` under various grid settings (*i.e.*, 9, 16, 25) to baselines (supervised and semi-supervised `ReCo`). Table 7 shows the mean IoU validation performance. We can see that for all cases, `ARCO-SG` and `ARCO-SAG` consistently improve performance, compared to `ReCo`, in all the semi-supervised settings. For example, under the fewest label setting in each dataset, compared to `ReCo`, applying stratified group sampling (SG) can improve results by an especially significant margin, with up to 2.4 – 7.8% relative gains. As shown in Pascal VOC 2012 [98] – 60 labels (Figure 10), 200 labels (Figure 11), 600 labels (Figure 12), Cityscapes [97] – 20 labels (Figure 13), 50 labels (Figure 14), 150 labels (Figure 15), SUN RGB-D [99] – 50 labels (Figure 16), 150 labels (Figure 17), 500 labels (Figure 18), `ARCO-SG` and `ARCO-SAG` consistently yield more accurate and sharper boundaries compared to `ReCo`. All those clearly demonstrate the superiority of our models.

## L  Generalization across Label Ratios and Frameworks

Besides generalizing well across different datasets and diverse labeled settings, we additionally evaluate the performance of SG and SAG sampling coupled with different CL frameworks (*i.e.*, `MoCov2` [103], $k$NN-`MoCo` [104], `SimCLR` [14], `BYOL` [26], `ISD` [27], `VICReg` [28]) on ACDC under

various label ratios (*i.e.*, 1%, 5%, 10%). In this work, we mainly study the state-of-the-art CL frameworks from the computer vision domain hereinafter for ablations, considering their superior performance in the task of image classification. Note that, for each fair comparison, we strictly follow the default setting in these CL frameworks [103, 104, 14, 26, 27, 28] for pretraining, and fine-tune each networks using the same settings in Appendix C. Full experimental details are in Appendix C.

Since they work orthogonally to our pretraining strategy, we conduct a comprehensive comparison of these CL-based frameworks in Table 8. Clearly, our proposed SG and SAG sampling help significantly in improving the segmentation performance across all the CL-based frameworks and is capable of being integrated with previous frameworks for further enhanced **model robustness**. Moreover, it is important to observe that the performance benefits of our methods increase significantly with a lower label setting. This observation augments the necessity of our proposed methods while training networks with high **label efficiency**.

## M  Final Checkpoint Loss Landscapes

From visualizations in Figure 19, `ARCO-SG` converges to much flatter loss valleys, which evidences their robustness in learning anatomical features.

## N  Ablation on Different Training Settings

**Hyperparameter Selection.**  For grid search, we detail the tuning steps here. The tuning is done in sequential order. $\lambda_1$ is chosen from $\{0.001, 0.005, 0.01, 0.05, 0.1, 1.0\}$, and $\lambda_2, \lambda_3$ are chosen from $\{0.1, 0.2, 0.5, 1.0, 2.0, 10.0\}$. We use the validation set to search over hyperparameters and find the best hyperparameter on ACDC with 1% labeled ratio. As shown in Figure 20, with a carefully tuned hyperparameters $\lambda_1$=0.01, $\lambda_2$=1.0, and $\lambda_3$=1.0, such setting achieves superior performance compared to others.

Table 6: Quantitative comparisons (DSC[%] ↑ / ASD[voxel] ↓) across the three labeled ratio settings (1%, 5%, 10%) on the MMWHS benchmark. All experiments are conducted as [30, 90, 91, 92, 93, 8, 94, 9, 57, 95, 62, 50, 59, 96, 53, 16, 17] in the identical setting for fair comparisons. Best and second-best results are coloured **blue** and red, respectively. UNet-F (fully-supervided) and UNet-L (semi-supervided) are considered as the upper bound and the lower bound for the performance comparison. Please refer to the text for discussion. Note that, Left Ventricle → LV, Right Ventricle → RV, Left Atrium → LA, Right Atrium → RA, Myocardium → Myo, Ascending Aorta → AA, Pulmonary Artery → PA. We adopt the identical data augmentation (*i.e.*, random rotation, random cropping, and horizontal flipping) for fair comparisons.

| Method (1% Labeled) | Average | LV | RV | LA | RA | Myo | AA | PA |
|---|---|---|---|---|---|---|---|---|
| UNet-F [30] | 85.8/8.01 | 87.0/4.11 | 79.5/14.7 | 92.7/4.96 | 81.6/13.1 | 83.9/9.32 | 95.0/3.33 | 81.1/6.46 |
| UNet-L | 58.3/33.9 | 73.5/28.2 | 57.9/37.6 | 74.9/30.2 | 47.2/65.4 | 61.9/27.8 | 74.0/18.7 | 18.6/29.4 |
| EM [90] | 59.5/63.2 | 71.7/53.0 | 64.7/26.3 | 66.6/51.5 | 61.7/39.4 | 56.8/66.0 | 81.9/48.8 | 12.9/157.2 |
| CCT [91] | 62.8/27.5 | 78.1/18.3 | 62.8/45.1 | 83.0/18.4 | 45.3/69.9 | 67.0/18.9 | 76.3/10.6 | 26.7/11.2 |
| DAN [92] | 63.8/39.0 | 76.3/15.4 | 62.2/58.3 | 68.0/25.3 | 52.3/48.2 | 57.0/54.3 | 89.0/28.0 | 42.0/43.4 |
| URPC [57] | 65.7/29.7 | 77.0/29.5 | 65.1/33.4 | 87.6/18.5 | 52.1/56.3 | 65.8/28.5 | 85.5/22.7 | 27.3/29.7 |
| DTC [9] | 62.9/32.3 | 78.0/29.6 | 63.6/29.4 | 74.6/30.8 | 49.4/53.5 | 67.3/28.1 | 89.0/10.0 | 18.2/44.4 |
| DCT [93] | 60.0/35.3 | 72.1/31.6 | 55.6/42.9 | 79.0/24.0 | 54.6/65.7 | 62.7/31.4 | 67.3/26.7 | 29.2/35.3 |
| ICT [95] | 59.9/32.8 | 77.1/15.5 | 53.4/41.8 | 79.7/25.4 | 44.1/69.0 | 62.9/28.5 | 74.1/20.8 | 28.4/28.8 |
| MT [59] | 61.3/36.0 | 70.7/37.5 | 62.8/29.0 | 74.0/49.6 | 52.8/58.5 | 58.7/27.0 | 85.3/13.0 | 24.9/37.5 |
| UAMT [8] | 61.1/37.6 | 75.1/31.9 | 60.3/49.2 | 79.5/30.7 | 50.8/81.8 | 62.5/33.1 | 76.6/11.7 | 23.0/25.0 |
| SASSNet [94] | 62.5/33.9 | 76.6/27.6 | 62.8/29.0 | 77.6/24.6 | 52.8/58.5 | 58.7/27.0 | 85.3/13.0 | 24.9/37.5 |
| CPS [62] | 69.2/22.6 | 77.9/20.1 | 67.9/25.3 | 87.8/18.0 | 67.6/27.7 | 66.7/14.2 | 91.5/20.3 | 24.8/32.5 |
| GCL [50] | 71.6/20.3 | 83.5/10.9 | 66.0/31.7 | 91.2/5.29 | 58.9/48.0 | 73.2/16.9 | 89.8/4.59 | 38.6/24.8 |
| MC-Net [96] | 65.4/56.9 | 67.6/59.0 | 59.6/45.8 | 68.8/29.6 | 61.7/62.8 | 53.8/56.8 | 84.4/80.5 | 61.9/63.6 |
| SS-Net [53] | 60.6/30.7 | 67.5/15.5 | 66.0/42.5 | 73.2/25.1 | 53.7/52.9 | 59.5/15.0 | 81.0/29.5 | 23.2/34.5 |
| ACTION [16] | 78.2/10.5 | 88.2/3.95 | 71.0/11.3 | 89.4/**3.00** | 74.1/32.5 | 79.0/8.52 | 83.2/3.82 | 62.3/10.2 |
| MONA [17] | 82.2/8.05 | 89.2/3.64 | 75.8/6.98 | 89.9/3.31 | 77.6/27.4 | 79.5/3.70 | 94.4/**2.72** | 69.4/8.61 |
| ● ARCO-SAG (ours) | 86.1/8.78 | 89.9/3.54 | 80.9/8.79 | 92.1/3.22 | 83.1/24.1 | 84.0/7.36 | 94.8/5.41 | 77.5/9.02 |
| ○ ARCO-SG (ours) | **87.3**/5.79 | **90.7**/3.43 | **82.5**/5.58 | **92.9**/3.13 | **86.2**/15.0 | **85.6**/2.89 | **95.2**/2.85 | **77.9**/7.62 |

| Method (5% Labeled) | Average | LV | RV | LA | RA | Myo | AA | PA |
|---|---|---|---|---|---|---|---|---|
| UNet-F [30] | 85.8/8.01 | 87.0/4.11 | 79.5/14.7 | 92.7/4.96 | 81.6/13.1 | 83.9/9.32 | 95.0/3.33 | 81.1/6.46 |
| UNet-L | 72.3/31.1 | 78.5/31.2 | 69.0/25.6 | 78.1/23.8 | 57.0/45.3 | 69.4/34.5 | 90.2/14.5 | 63.9/42.5 |
| EM [90] | 80.6/17.3 | 82.0/22.7 | 75.3/26.5 | 87.9/19.2 | 72.8/19.4 | 74.8/19.1 | 94.1/6.52 | 77.6/7.68 |
| CCT [91] | 79.0/21.9 | 82.9/15.3 | 73.5/31.6 | 83.3/14.4 | 75.5/34.0 | 77.9/24.3 | 92.7/17.6 | 67.0/16.1 |
| DAN [92] | 79.4/22.7 | 80.1/37.0 | 77.2/30.0 | 83.1/12.3 | 74.4/13.9 | 78.9/27.4 | 92.2/28.9 | 69.5/22.7 |
| URPC [57] | 76.3/25.5 | 84.2/18.8 | 71.3/25.2 | 78.5/18.2 | 63.4/39.4 | 75.5/17.3 | 93.5/17.3 | 71.0/34.9 |
| DTC [9] | 76.4/21.3 | 82.3/17.2 | 72.4/24.4 | 76.1/15.2 | 65.0/31.8 | 75.2/20.9 | 92.8/13.1 | 70.8/26.7 |
| DCT [93] | 80.8/23.0 | 84.0/45.4 | 75.7/26.0 | 87.9/12.1 | 73.9/31.7 | 77.2/34.1 | 94.6/5.26 | 72.5/6.45 |
| ICT [95] | 77.9/18.8 | 84.1/13.2 | 76.7/26.3 | 79.2/14.0 | 66.5/24.9 | 74.1/18.8 | 94.2/6.21 | 70.3/28.3 |
| MT [59] | 77.5/24.2 | 83.5/15.4 | 72.8/29.0 | 78.0/16.2 | 68.9/39.2 | 74.7/24.4 | 93.3/11.8 | 71.1/33.4 |
| UAMT [8] | 76.2/21.1 | 83.4/20.7 | 71.5/24.4 | 77.0/14.6 | 62.8/30.6 | 75.8/22.1 | 93.0/8.91 | 69.7/26.0 |
| SASSNet [94] | 75.2/25.0 | 80.9/25.0 | 70.8/31.2 | 80.0/17.0 | 61.4/40.0 | 70.0/31.3 | 92.4/18.9 | 71.0/21.9 |
| CPS [62] | 78.3/22.5 | 83.0/29.6 | 68.8/27.7 | 85.0/20.4 | 73.1/15.5 | 71.9/35.2 | 94.7/9.05 | 71.9/20.2 |
| GCL [50] | 83.5/7.41 | 86.4/4.72 | 78.5/9.79 | 88.6/4.34 | 79.8/12.1 | 81.4/8.07 | 93.5/3.91 | 76.4/8.95 |
| MC-Net [96] | 78.5/23.9 | 83.7/23.8 | 74.4/25.7 | 81.9/16.8 | 70.8/38.4 | 74.4/23.9 | 93.3/14.2 | 71.1/25.0 |
| SS-Net [53] | 78.0/25.2 | 83.0/7.76 | 74.8/32.0 | 82.3/20.3 | 69.6/42.6 | 71.1/15.8 | 92.4/25.5 | 73.2/32.1 |
| ACTION [16] | 85.4/6.71 | 88.2/3.09 | 78.8/9.66 | 90.5/2.84 | 80.6/15.6 | 84.4/7.37 | 94.0/2.56 | 81.3/5.86 |
| MONA [17] | 87.3/6.62 | 90.2/2.92 | 80.9/9.62 | 92.8/2.65 | 82.5/15.2 | 87.0/8.53 | 95.3/1.86 | 82.7/5.60 |
| ● ARCO-SAG (ours) | 88.6/6.73 | 91.1/**2.70** | 83.4/13.0 | 92.9/2.84 | 84.3/16.1 | 89.0/4.68 | 95.9/1.57 | 83.3/6.19 |
| ○ ARCO-SG (ours) | **89.3**/4.80 | **91.2**/2.70 | **84.6**/8.30 | **93.7**/2.49 | **85.6**/10.4 | **89.2**/3.41 | **96.0**/1.42 | **84.7**/4.95 |

| Method (10% Labeled) | Average | LV | RV | LA | RA | Myo | AA | PA |
|---|---|---|---|---|---|---|---|---|
| UNet-F [30] | 85.8/8.01 | 87.0/4.11 | 79.5/14.7 | 92.7/4.96 | 81.6/13.1 | 83.9/9.32 | 95.0/3.33 | 81.1/6.46 |
| UNet-L | 77.8/19.7 | 82.8/8.92 | 77.3/16.1 | 75.9/22.8 | 74.6/24.7 | 75.3/11.6 | 90.8/21.6 | 67.8/32.3 |
| EM [90] | 82.1/15.1 | 86.7/19.8 | 78.4/24.5 | 88.1/7.46 | 77.6/15.9 | 75.8/25.1 | 95.0/4.13 | 73.2/8.86 |
| CCT [91] | 79.4/16.3 | 85.4/5.65 | 73.5/30.0 | 89.1/7.10 | 68.2/31.2 | 70.2/24.3 | 92.4/5.90 | 77.1/9.77 |
| DAN [92] | 80.2/15.6 | 81.6/22.6 | 74.2/21.2 | 88.0/10.1 | 75.5/17.1 | 76.9/20.2 | 94.0/4.40 | 74.4/9.15 |
| URPC [57] | 81.9/12.3 | 88.1/9.41 | 68.3/20.7 | 88.1/6.73 | 76.6/14.3 | 80.4/19.6 | 94.5/4.26 | 77.2/11.0 |
| DTC [9] | 79.5/20.6 | 82.8/10.8 | 75.8/18.7 | 85.9/13.8 | 75.2/41.5 | 74.4/13.9 | 90.7/25.1 | 71.7/20.1 |
| DCT [93] | 82.8/12.5 | 85.4/11.6 | 78.0/23.3 | 89.0/4.30 | 79.0/16.5 | 75.5/19.1 | 94.3/4.42 | 78.4/8.08 |
| ICT [95] | 82.2/12.0 | 88.4/5.11 | 75.0/13.5 | 89.0/6.98 | 75.2/26.4 | 79.6/20.4 | 94.9/4.29 | 73.3/7.30 |
| MT [59] | 79.4/19.8 | 80.4/24.1 | 70.3/21.3 | 86.0/18.0 | 80.0/17.0 | 73.3/28.7 | 92.3/20.9 | 73.8/8.92 |
| UAMT [8] | 83.7/14.2 | 86.7/12.3 | 80.3/20.6 | 89.6/8.10 | 79.5/19.2 | 79.2/19.6 | 93.9/10.3 | 73.8/8.92 |
| SASSNet [94] | 81.8/15.5 | 84.9/8.01 | 78.3/15.9 | 84.4/12.5 | 79.3/27.3 | 79.0/14.6 | 93.4/8.30 | 73.3/22.3 |
| CPS [62] | 82.0/13.1 | 84.4/9.85 | 78.5/21.1 | 85.9/6.61 | 81.0/18.7 | 76.4/18.3 | 93.2/7.04 | 74.9/10.3 |
| GCL [50] | 86.7/8.76 | 90.5/2.95 | 81.3/19.6 | 90.4/4.31 | 83.1/18.1 | 86.7/5.84 | 94.8/2.00 | 80.3/8.53 |
| MC-Net [96] | 81.9/15.4 | 85.4/5.78 | 80.1/17.2 | 81.5/11.1 | 79.7/34.1 | 79.8/10.9 | 93.1/6.28 | 73.7/22.4 |
| SS-Net [53] | 82.3/13.9 | 85.7/8.80 | 79.5/17.6 | 84.1/12.1 | 80.2/20.0 | 81.0/14.0 | 93.6/8.60 | 72.0/16.1 |
| ACTION [16] | 86.1/5.93 | 88.9/3.25 | 81.3/6.99 | 89.4/3.13 | 81.6/14.1 | 87.3/3.76 | 94.4/2.53 | 79.4/7.78 |
| MONA [17] | 87.6/6.83 | 90.7/2.89 | 82.8/8.99 | 91.8/3.48 | 85.2/15.7 | 87.2/5.32 | 94.9/4.32 | 80.4/7.13 |
| ● ARCO-SAG (ours) | 89.3/4.42 | 91.1/2.97 | 84.8/6.30 | 94.1/2.38 | 86.1/9.73 | 89.2/2.89 | 96.0/1.68 | 83.6/5.02 |
| ○ ARCO-SG (ours) | **89.4**/4.80 | **91.6**/2.56 | **85.0**/6.79 | 93.9/2.53 | **86.3**/11.5 | **89.6**/2.71 | 95.9/1.66 | 83.6/5.88 |

Table 7: Quantitative comparisons (Intersection Over Union (IoU) ↑) for Pascal VOC, CityScapes and SUN RGB-D datasets. All experiments are conducted as [100] in the identical setting for fair comparisons. Best and second-best results are colored **blue** and red, respectively.

| Method | Pascal VOC | | | | CityScapes | | | | SUN RGB-D | | | |
|---|---|---|---|---|---|---|---|---|---|---|---|---|
| | 60 labels | 200 labels | 600 labels | all labels | 20 labels | 50 labels | 150 labels | all labels | 50 labels | 150 labels | 500 labels | all labels |
| Supervised | 39.4 | 55.5 | 64.6 | 77.8 | 38.2 | 45.9 | 55.4 | 70.9 | 20.0 | 29.2 | 38.9 | 51.8 |
| ReCo [100] + ClassMix | 57.1 | 69.4 | 73.2 | - | 49.9 | 57.9 | 65.0 | - | 30.5 | 40.4 | 44.6 | - |
| ● ARCO-SAG (9 Grid) + ClassMix | 58.3 | 70.5 | 75.4 | - | 50.2 | 60.2 | 66.5 | - | 31.5 | 40.9 | 45.7 | - |
| ● ARCO-SAG (16 Grid) + ClassMix | 58.7 | 70.9 | 75.1 | - | 50.1 | 60.6 | 66.3 | - | 37.8 | 40.2 | 45.7 | - |
| ● ARCO-SAG (25 Grid) + ClassMix | 59.1 | 70.9 | 74.9 | - | 49.8 | 60.6 | 66.7 | - | 38.5 | 40.5 | 45.5 | - |
| ○ ARCO-SG (9 Grid) + ClassMix | 59.2 | 71.8 | 75.3 | - | 52.5 | 60.9 | 66.8 | - | 32.4 | 41.4 | 46.6 | - |
| ○ ARCO-SG (16 Grid) + ClassMix | 59.6 | 71.7 | 75.5 | - | 53.7 | 61.2 | 66.2 | - | 37.7 | 41.0 | 46.4 | - |
| ○ ARCO-SG (25 Grid) + ClassMix | 59.5 | 71.7 | 75.2 | - | 51.5 | 61.8 | 66.4 | - | 38.3 | 41.5 | 47.3 | - |

Table 8: Ablation on different contrastive learning frameworks with respect to various labeled ratio settings (1%, 5%, 10%). Experiments are conducted on ACDC using UNet [30] as the backbone. Here we report the segmentation performance in terms of DSC[%] and ASD[mm]. On all three labeled settings, incorporating our methods (*i.e.*, SG and SAG) consistently achieve superior model robustness gains compared to naïve sampling across different state-of-the-art CL frameworks.

| Category | Method | 1% Labeled | | 5% Labeled | | 10% Labeled | |
|---|---|---|---|---|---|---|---|
| | | DSC ↑ | ASD ↓ | DSC ↑ | ASD ↓ | DSC ↑ | ASD ↓ |
| MoCov2 [103] | NS | 76,4 | 5.64 | 83.3 | 3.78 | 83.8 | 3.17 |
| | ● SAG | 80.9 | 4.16 | 84.7 | 3.53 | 84.6 | 3.84 |
| | ○ SG | 81.4 | 4.01 | 85.1 | 1.48 | 84.9 | 3.08 |
| $k$NN-MoCo [104] | NS | 78.3 | 4.54 | 83.8 | 3.74 | 84.1 | 2.97 |
| | ● SAG | 81.8 | 4.08 | 84.6 | 3.41 | 84.7 | 2.58 |
| | ○ SG | 83.9 | 3.17 | 85.4 | 3.17 | 85.0 | 2.47 |
| SimCLR [14] | NS | 74.9 | 4.89 | 80.9 | 3.19 | 84.1 | 2.78 |
| | ● SAG | 78.5 | 4.01 | 83.8 | 2.68 | 85.8 | 2.01 |
| | ○ SG | 79.1 | 3.49 | 84.3 | 2.31 | 86.0 | 1.76 |
| BYOL [26] | NS | 77.3 | 5.01 | 82.9 | 3.14 | 84.8 | 1.67 |
| | ● SAG | 79.8 | 4.12 | 85.0 | 2.81 | 85.7 | 1.36 |
| | ○ SG | 80.2 | 3.79 | 85.8 | 1.67 | 85.9 | 1.13 |
| ISD [27] | NS | 79.3 | 3.54 | 81.2 | 2.86 | 85.6 | 1.90 |
| | ● SAG | 81.4 | 3.50 | 83.1 | 2.53 | 86.3 | 1.61 |
| | ○ SG | 82.2 | 2.04 | 83.7 | 1.97 | 86.7 | 1.34 |
| VICReg [28] | NS | 64.0 | 10.6 | 79.1 | 4.18 | 82.9 | 3.89 |
| | ● SAG | 81.1 | 3.49 | 83.8 | 3.02 | 86.3 | 2.14 |
| | ○ SG | 81.6 | 3.12 | 84.1 | 2.78 | 86.8 | 2.01 |
| ARCO (ours) | NS | 82.6 | 1.43 | 86.9 | 1.07 | 87.7 | 1.33 |
| | ● SAG | 84.9 | 1.47 | 87.1 | 0.848 | 88.5 | 1.40 |
| | ○ SG | 85.5 | 0.947 | 88.7 | 0.841 | 89.4 | 0.776 |

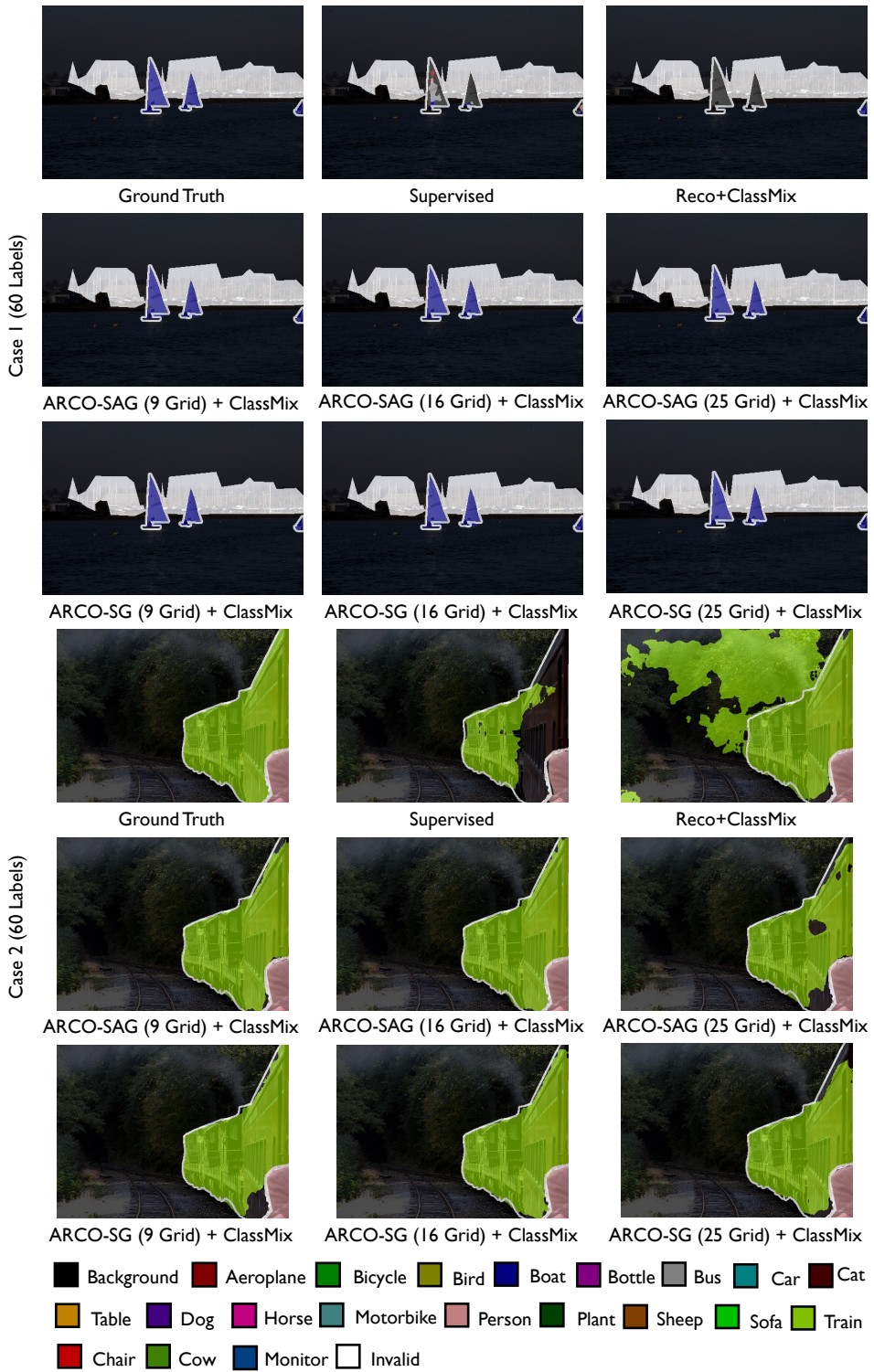

Figure 10: Visual results on Pascal validation set with 60 labels. As is shown, `ARCO-SG` and `ARCO-SAG` consistently yield more accurate and sharper boundaries compared to all other SSL methods.

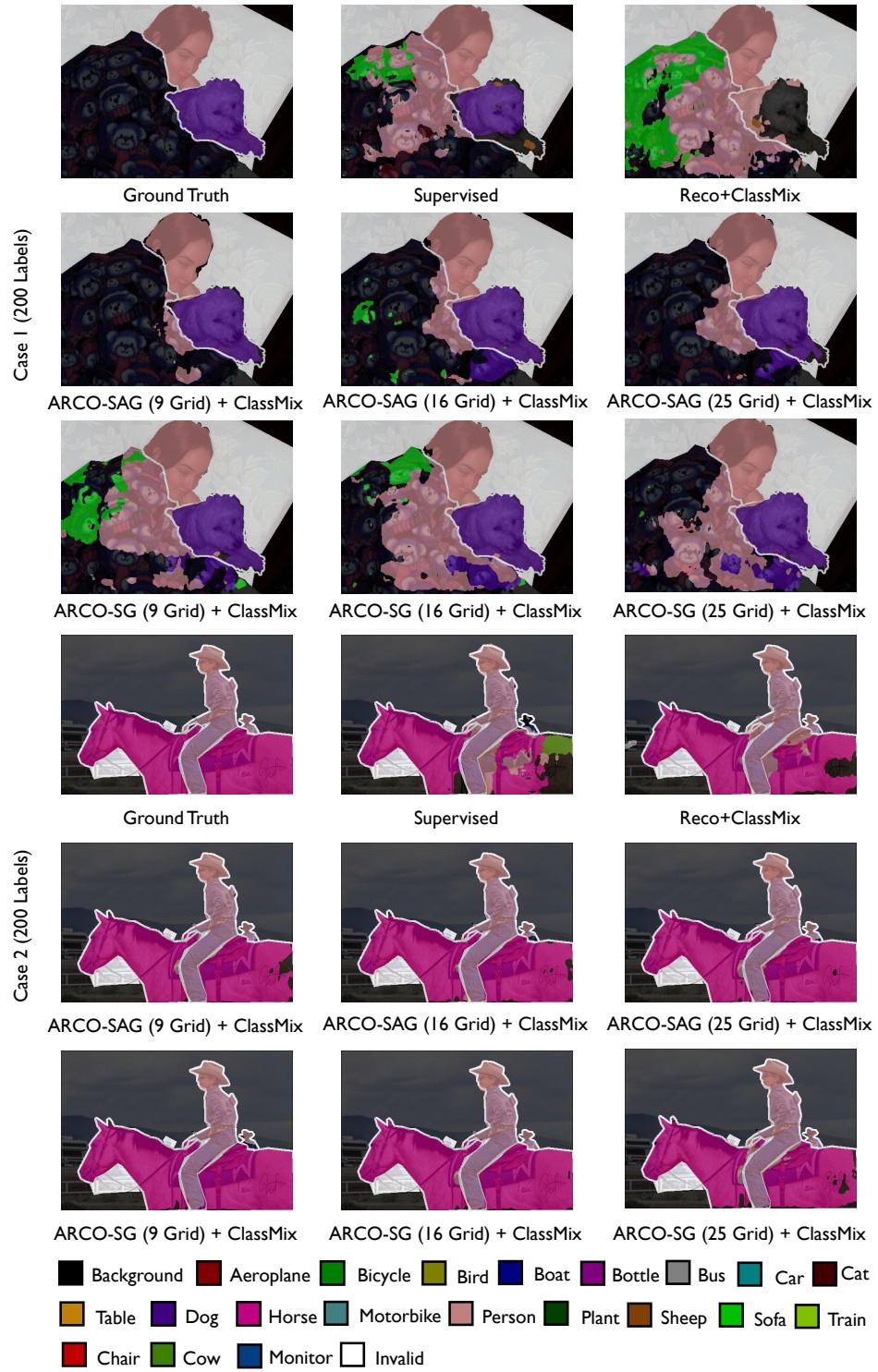

Figure 11: Visual results on Pascal validation set with 200 labels. As is shown, `ARCO-SG` and `ARCO-SAG` consistently yield more accurate and sharper boundaries compared to all other SSL methods.

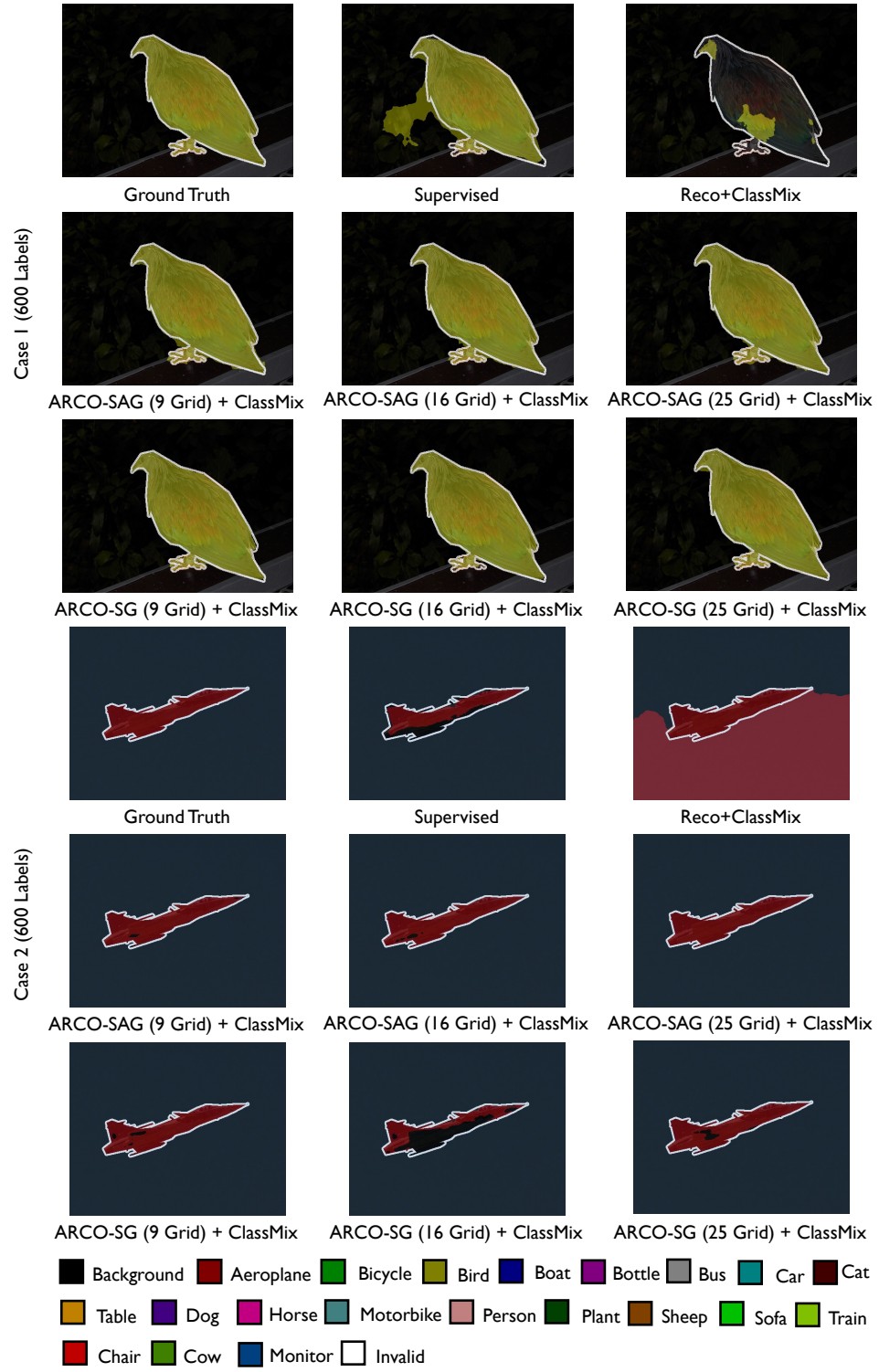

Figure 12: Visual results on Pascal validation set with 600 labels. As is shown, `ARCO-SG` and `ARCO-SAG` consistently yield more accurate and sharper boundaries compared to all other SSL methods.

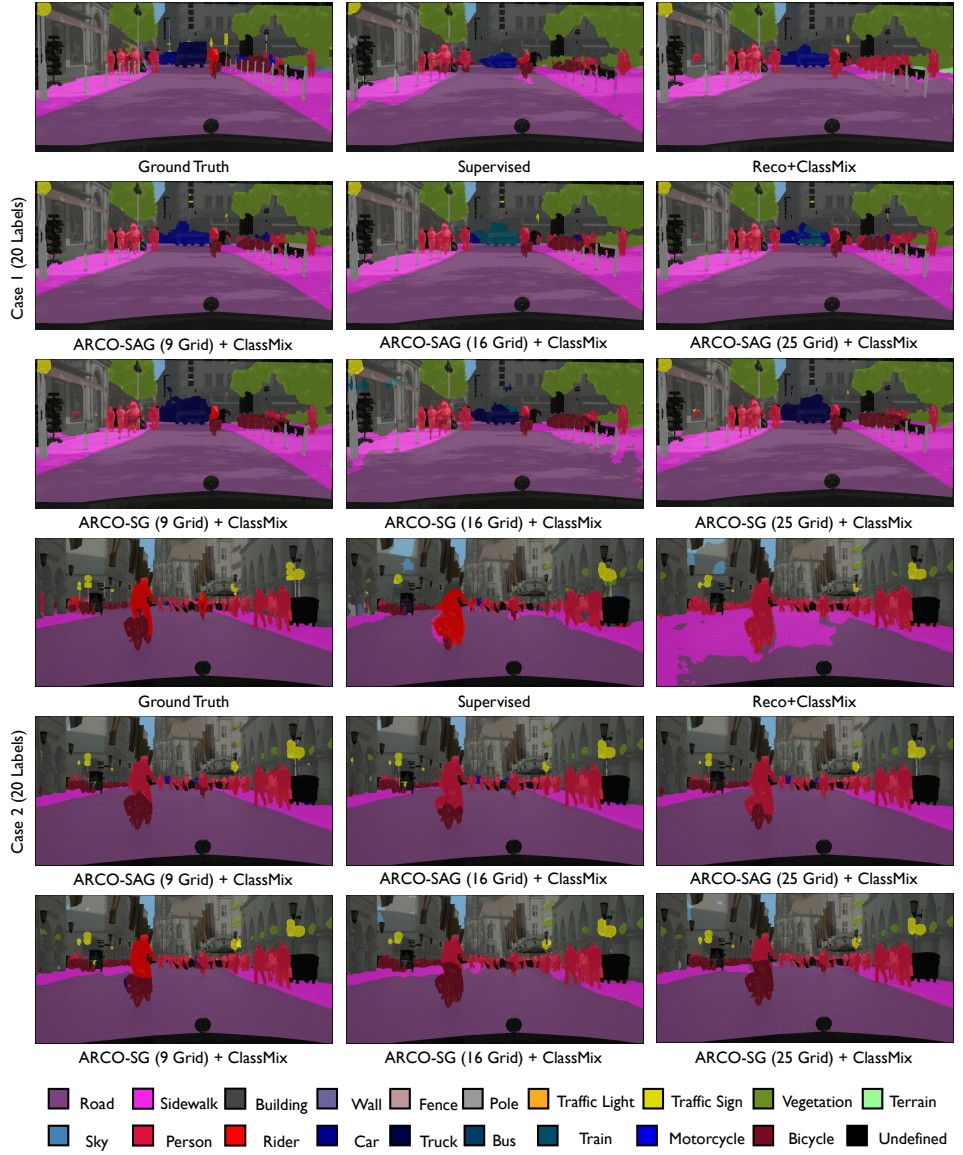

Figure 13: Visual results on Cityscapes validation set with 20 labels. As is shown, `ARCO-SG` and `ARCO-SAG` consistently yield more accurate and sharper boundaries compared to all other SSL methods.

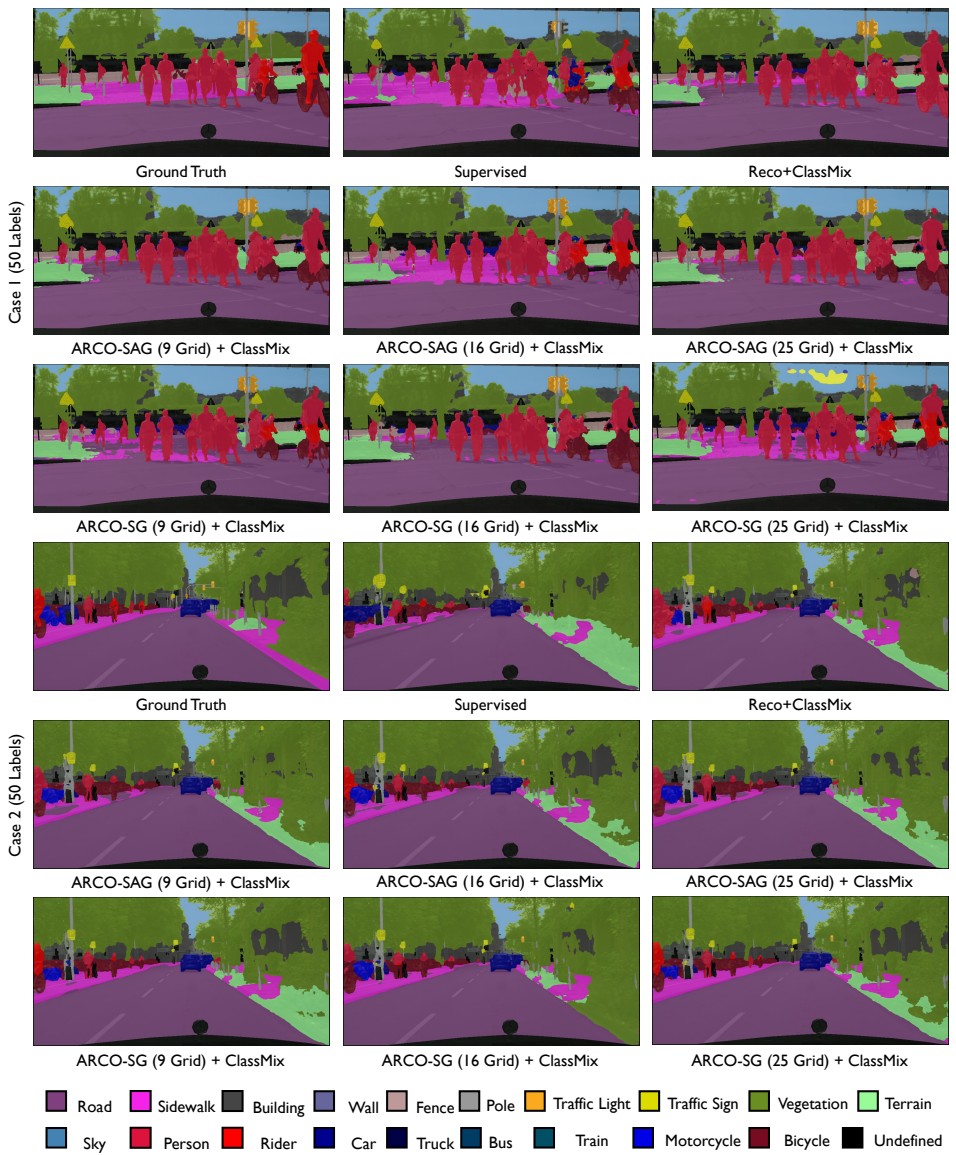

Figure 14: Visual results on Cityscapes validation set with 50 labels. As is shown, `ARCO-SG` and `ARCO-SAG` consistently yield more accurate and sharper boundaries compared to all other SSL methods.

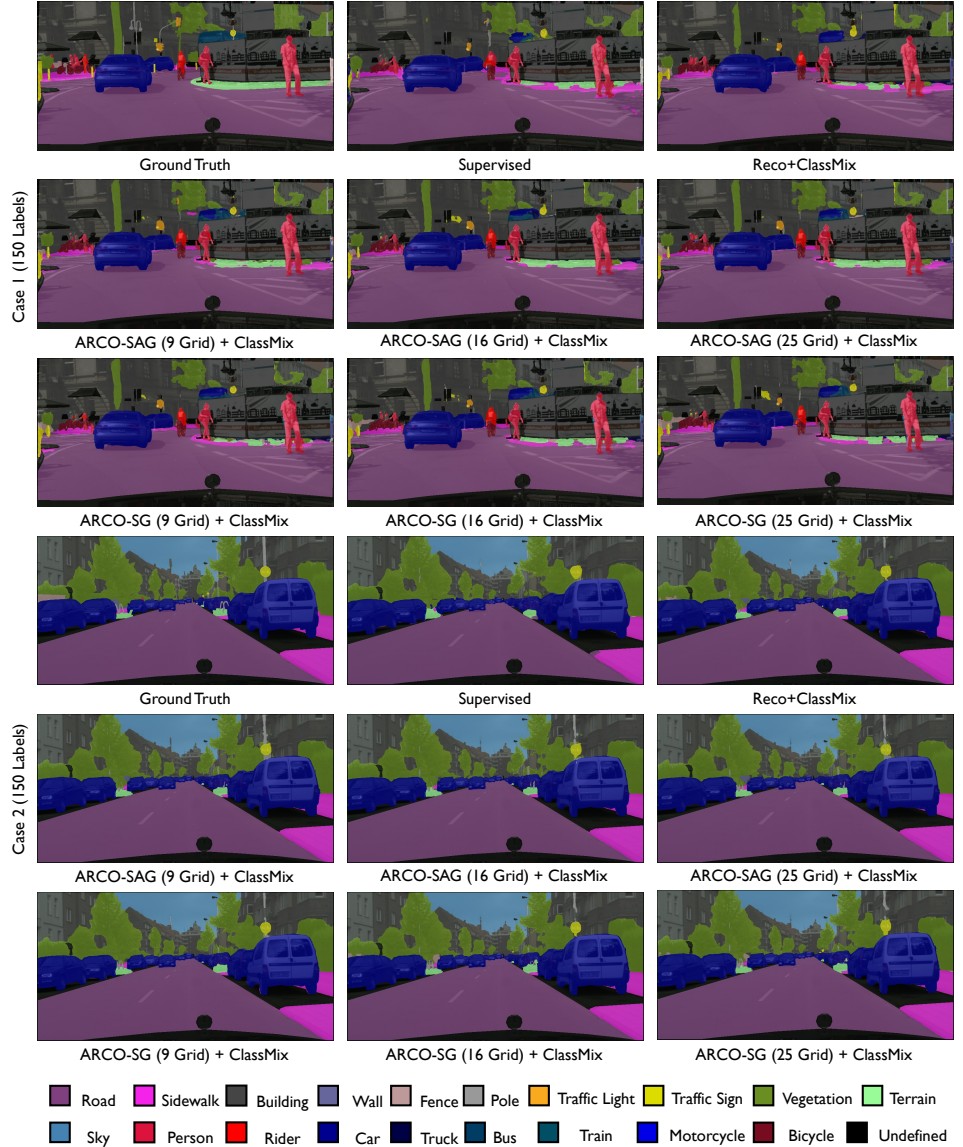

Figure 15: Visual results on Cityscapes validation set with 150 labels. As is shown, `ARCO-SG` and `ARCO-SAG` consistently yield more accurate and sharper boundaries compared to all other SSL methods.

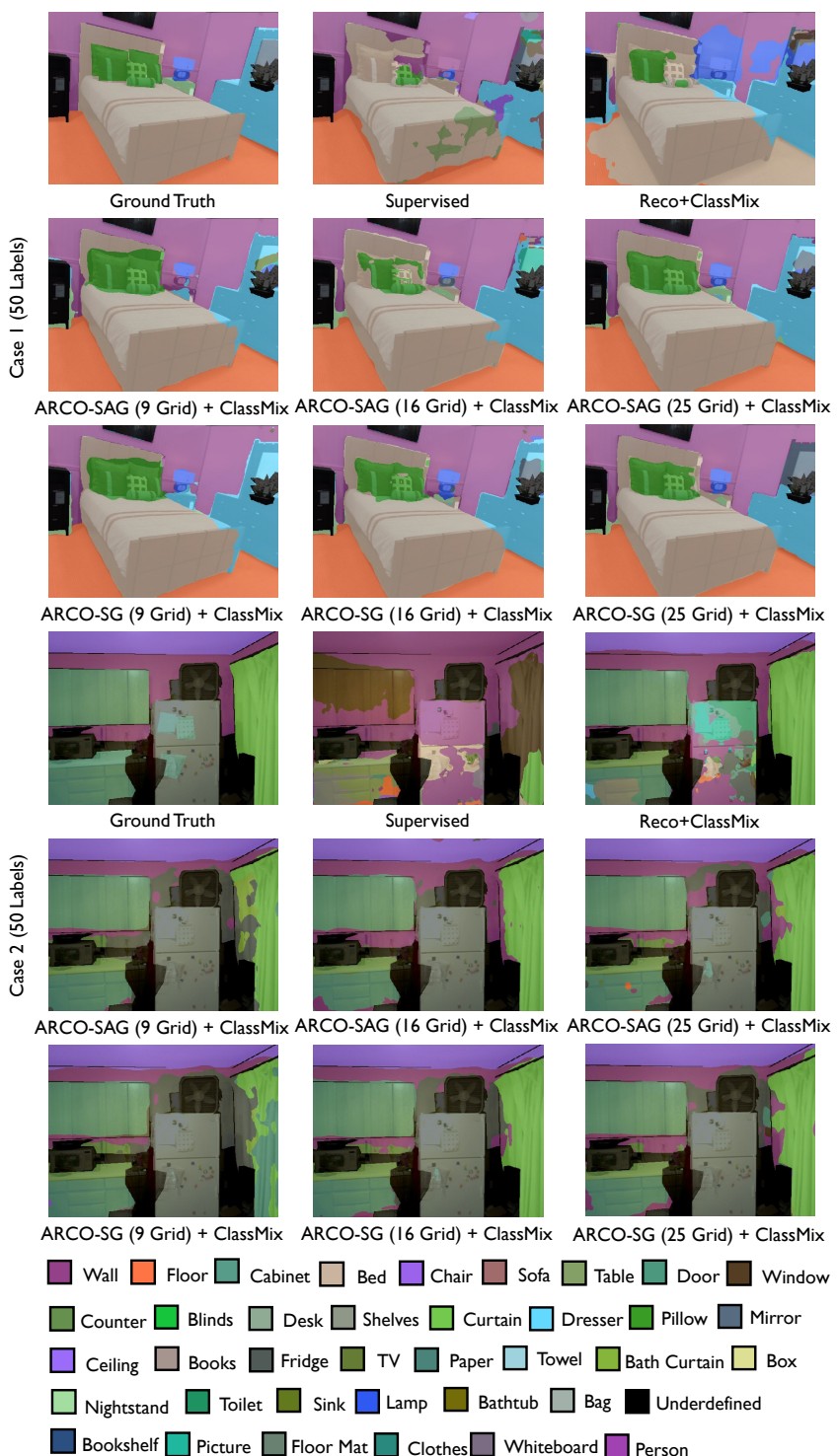

Figure 16: Visual results on SUN RGB-D validation set with 50 labels. As is shown, `ARCO-SG` and `ARCO-SAG` consistently yield more accurate and sharper boundaries compared to all other SSL methods.

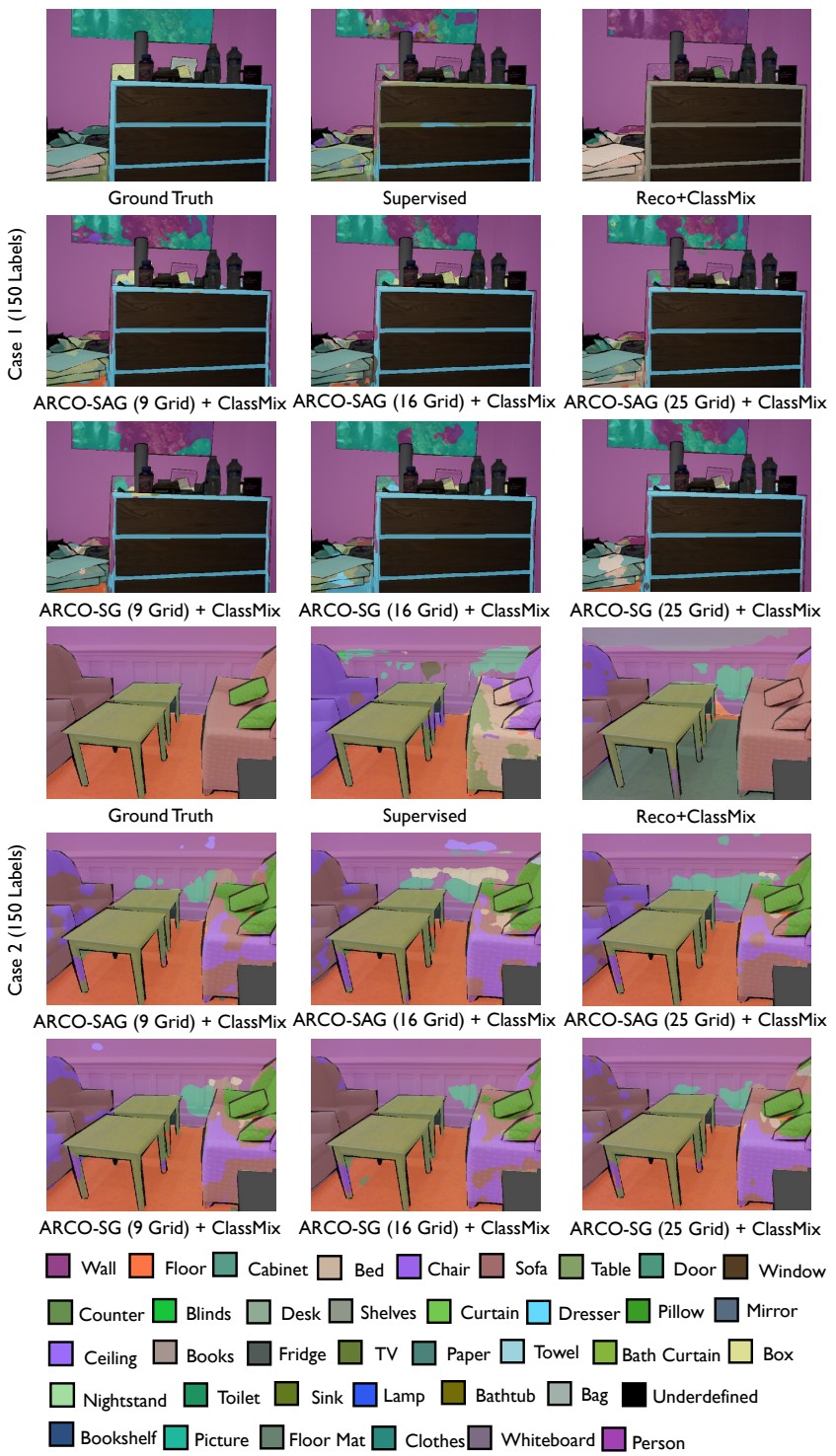

Figure 17: Visual results on SUN RGB-D validation set with 150 labels. As is shown, `ARCO-SG` and `ARCO-SAG` consistently yield more accurate and sharper boundaries compared to all other SSL methods.

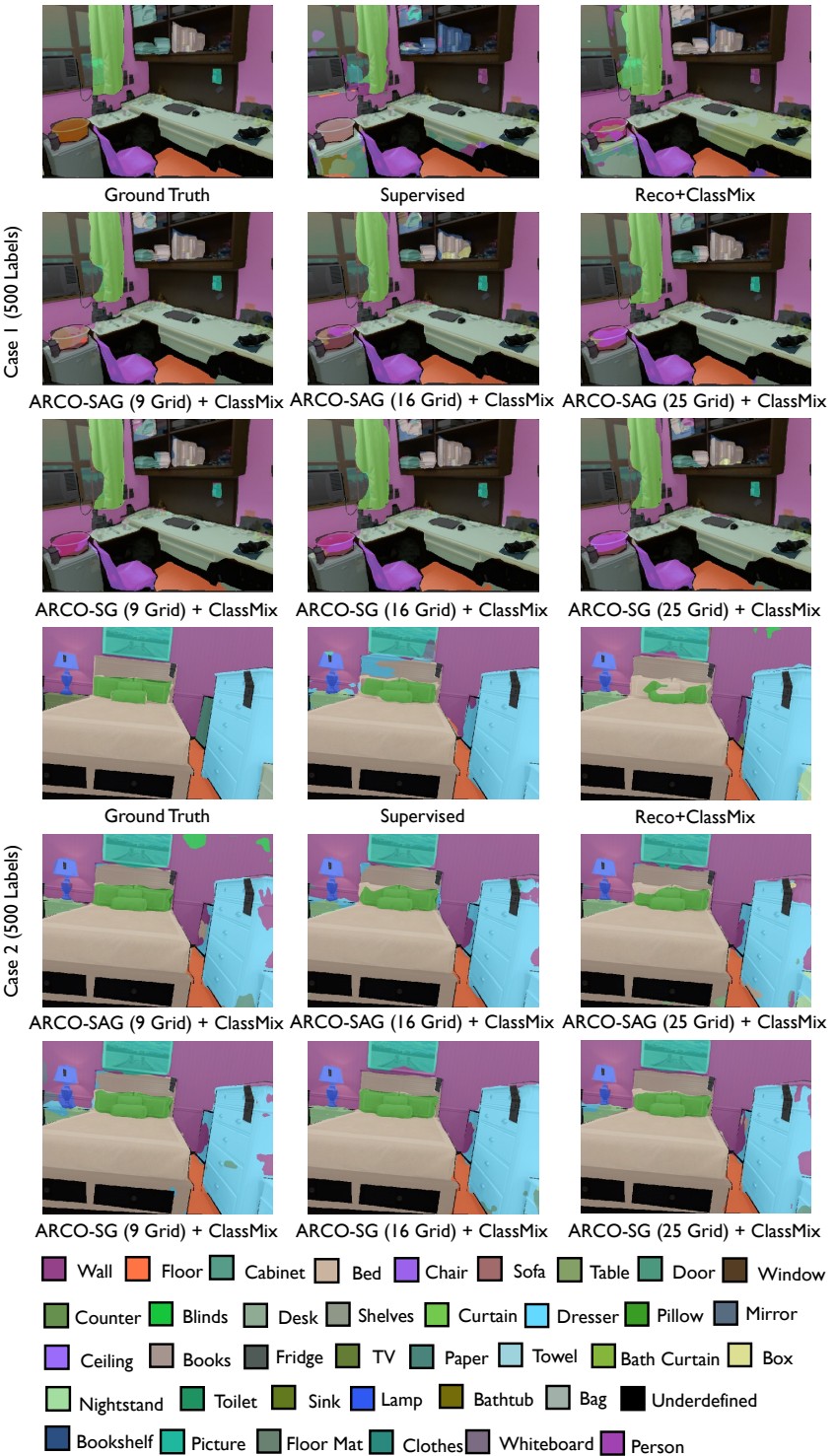

Figure 18: Visual results on SUN RGB-D validation set with 500 labels. As is shown, `ARCO-SG` and `ARCO-SAG` consistently yield more accurate and sharper boundaries compared to all other SSL methods.

Table 9: Quantitative comparisons (DSC[%] ↑ / ASD[voxel] ↓) across the three labeled ratio settings (1%, 5%, 10%) on the 3D MP-MRI and 3D LA benchmarks. All 3D experiments are conducted as [78, 90, 91, 92, 93, 8, 94, 9, 57, 95, 62, 50, 59, 96, 53, 16, 17] in the identical setting for fair comparisons. Best and second-best results are coloured **blue** and red, respectively. `VNet-F` (fully-supervised) and `VNet-L` (semi-supervided) are considered as the upper bound and the lower bound for the performance comparison. Please refer to the text for discussion.

| | MP-MRI | | | LA | | |
|---|---|---|---|---|---|---|
| | 1% Labeled | 5% Labeled | 10% Labeled | 1 Labeled (1%) | 4 Labeled (5%) | 8 Labeled (10%) |
| 3D Method | Liver | Liver | Liver | Left Atrium (LA) | Left Atrium (LA) | Left Atrium (LA) |
| VNet-F [78] | 93.1/5.73 | 93.1/5.73 | 93.1/5.73 | 91.5/1.51 | 91.5/1.51 | 91.5/1.51 |
| VNet-L | 68.6/33.4 | 81.6/12.5 | 87.9/7.55 | 40.0/21.2 | 52.6/9.87 | 82.7/3.26 |
| EM [90] | 73.2/30.1 | 86.0/15.8 | 91.9/6.89 | 48.3/21.3 | 81.1/4.68 | 82.7/4.77 |
| CCT [91] | 74.2/24.5 | 86.0/15.8 | 90.9/9.54 | 40.3/13.8 | 70.8/8.31 | 82.0/5.25 |
| DAN [92] | 69.4/31.4 | 88.3/10.2 | 91.1/8.76 | 38.5/22.0 | 78.8/6.53 | 80.2/5.37 |
| URPC [57] | 72.7/29.9 | 89.8/9.00 | 91.7/7.41 | 65.0/8.97 | 80.2/5.48 | 83.1/4.57 |
| DTC [9] | 78.6/18.8 | 89.6/10.1 | 90.5/11.2 | 36.2/11.7 | 83.6/2.81 | 87.1/2.23 |
| DCT [93] | 74.1/31.1 | 88.2/12.4 | 90.1/11.1 | 42.9/19.1 | 80.1/9.06 | 80.4/9.18 |
| ICT [95] | 72.3/30.5 | 88.6/11.2 | 91.1/8.46 | 47.7/16.0 | 78.4/6.96 | 85.4/4.14 |
| MT [59] | 73.8/29.4 | 87.7/12.8 | 92.0/7.15 | 58.1/17.8 | 77.0/8.15 | 82.8/5.90 |
| UAMT [8] | 71.6/31.2 | 87.1/12.8 | 91.3/9.71 | 60.3/11.3 | 82.3/3.82 | 87.8/2.12 |
| SASSNet [94] | 78.8/19.6 | 88.4/13.1 | 88.7/13.1 | 51.5/14.6 | 81.6/3.58 | 87.5/2.59 |
| CPS [62] | 80.0/17.1 | 89.2/10.8 | 91.0/9.16 | 45.1/22.0 | 79.7/9.28 | 80.7/5.16 |
| GCL [50] | 78.9/14.2 | 87.9/8.29 | 90.4/5.68 | 52.6/12.8 | 75.5/7.60 | 84.8/4.22 |
| MC-Net [96] | 79.7/20.2 | 90.1/8.27 | 92.4/**4.34** | 44.3/14.1 | 83.6/2.70 | 87.6/1.82 |
| SS-Net [53] | 88.0/8.93 | 90.9/9.94 | 92.0/5.67 | 43.4/14.8 | 86.3/2.31 | 88.6/1.90 |
| ACTION [16] | 86.5/13.6 | 90.3/12.3 | 90.9/10.0 | 71.1/6.23 | 86.6/2.24 | 88.7/2.10 |
| MONA [17] | 91.3/5.31 | 92.2/9.46 | 92.3/8.16 | 72.8/10.7 | 87.0/2.81 | 89.5/2.40 |
| • ARCO-SAG (ours) | 91.5/6.82 | **92.5**/6.95 | 92.6/7.54 | 73.2/6.47 | 86.9/2.73 | 89.1/2.30 |
| ∘ ARCO-SG (ours) | **91.6**/6.60 | **92.5**/**6.31** | **92.8**/6.64 | **75.0**/**4.06** | **87.8**/**1.66** | **89.9**/**1.47** |

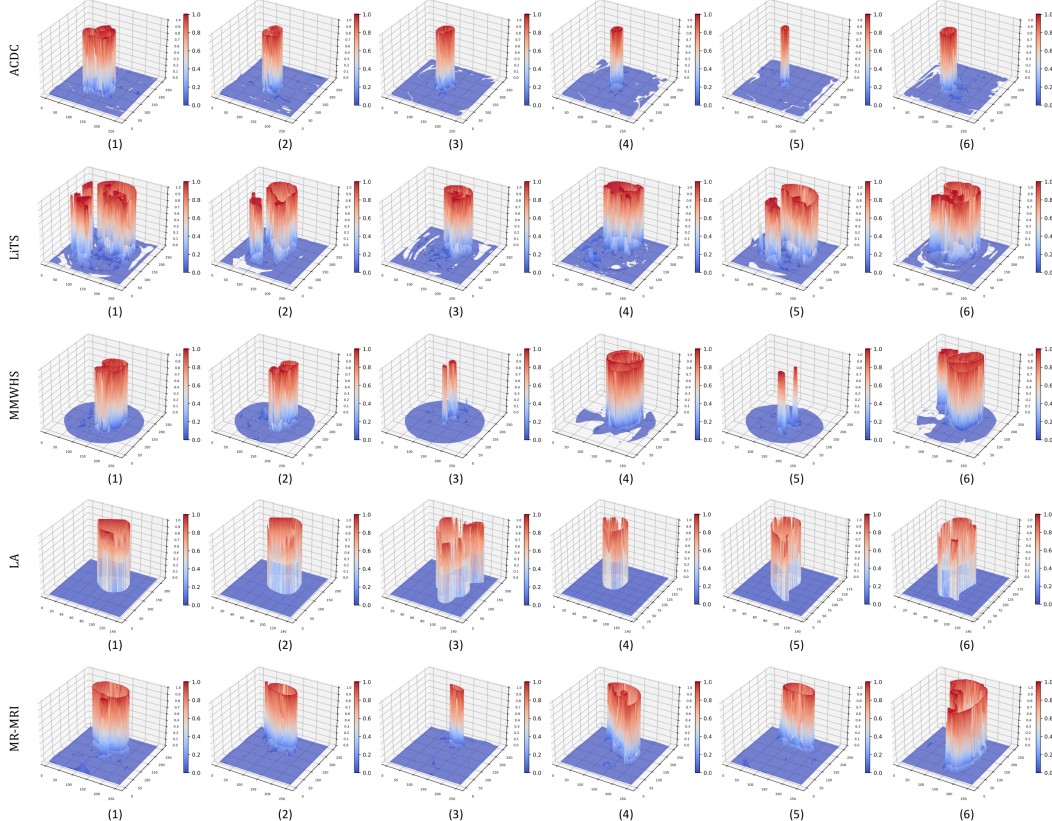

Figure 19: Loss landscape visualization of pixel-wise contrastive loss $\mathcal{L}_{\text{contrast}}$ with `ARCO-SG`. Loss plots are generated with same original images randomly chosen from ACDC [81], LiTS [82], MMWHS [83], LA [84], and MP-MRI, respectively. $z$-axis denotes the loss value at each pixel.

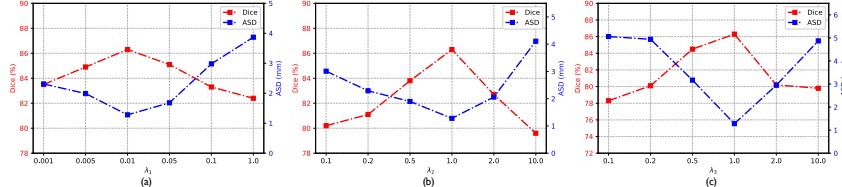

Figure 20: Effects of hyperparameters $\lambda_1, \lambda_2, \lambda_3$. We report Dice and ASD of `ARCO-SG` on ACDC with 1% labeled ratio.

