# Appendix to
# Rethinking Semi-Supervised Medical Image Segmentation: A Variance-Reduction Perspective

## A Theoretical Results Details

### A.1 Proof of Lemma 3.1

*Proof.* By definition of the estimate $\widehat{H}$, since the sampling from each group is independent, the variance can be written as

$$\text{Var}[\widehat{H}_{\text{SG}}] = \frac{1}{n^2} \sum_{m=1}^{M} \sum_{p \in \mathcal{D}_m} \sigma_m^2.$$

Since $|\mathcal{D}_m| = n_m$, it follows that

$$\text{Var}[\widehat{H}_{\text{SG}}] = \sum_{m=1}^{M} \frac{n_m^2}{n^2} \frac{\sigma_m^2}{n_m}.$$

Note that $\sigma_m^2$ is the variance of the $h(\mathbf{x}; p)$ where $p \sim \mathcal{P}_m$ uniformly.

**Analysis of SAG.** We now consider SAG. Recall that, compared to SG, SAG first samples $n_m/2$ pixels from $\mathcal{P}_m$[6] from each $m = 1, \cdots, M$ in the same way as SG. SAG then deterministically picks the rest $n_m/2$ pixels to be the reflection pixels of the first half (see Figure 2(3)). Let's choose arbitrary group $m$ and denote by $p$ and $p'$ the sampled pixel and its reflection from that group. Note that the variance of $p$ satisfies $\text{Var}_{\text{SAG}}[h(\mathbf{x}; p)] = \sigma_m^2$ since $p$ is sampled in a same way as in the case of SG. Observe that, by symmetry, the variance of $p$ and $p'$ should be the same $\text{Var}_{\text{SAG}}[h(\mathbf{x}; p)] = \text{Var}_{\text{SAG}}[h(\mathbf{x}; p)] = \sigma_m^2$. It follows that:

$$\begin{aligned}
&\text{Var}_{\text{SAG}}[h(\mathbf{x}; p) + h(\mathbf{x}; p')] \\
&= \text{Var}_{\text{SAG}}[h(\mathbf{x}; p)] + \text{Var}_{\text{SAG}}[h(\mathbf{x}; p')] + 2\text{Cov}_{\text{