# OpenReview forum: "Rethinking Semi-Supervised Medical Image Segmentation: A Variance-Reduction Perspective"
_NeurIPS.cc/2023/Conference — NeurIPS 2023 poster_

### Official Review · Reviewer_SMeb · 2023-07-01

**Soundness:** 3 good
**Presentation:** 2 fair
**Contribution:** 3 good
**Rating:** 7
**Confidence:** 3

**Summary:**

Edit: Updating score from 6 t o7 based on the discussions.

Extracting useful representations from unlabelled data for label efficient training of medical image segmentation task is a widely studied problem. This work approaches learning useful representations using contrastive learning (CL) within a semi-supervised setting. Strategies for obtaining variance reducing pixel partitions for CL are presented, along with theoretical analysis that show their variance reduction properties. The variance reduction estimation is also used to improve training stability and convergence. Comprehensive experiments on multiple medical imaging and computer vision datasets are performed showing strong performance improvements compared to other SSL methods. The authors show their method is label efficient across all the datasets.

**Strengths:**

* Focusing on variance reduction guarantees to extract contrastive samples for pixel level training is a strong contribution of this work.

* The theoretical analyses showing their unbiasedness and use of variance reduction techniques to improve training stability can have important applications in related domains like self-supervised learning.

* The  experimental evaluation is extensive, with strong performance improvements on multiple datasets.

* Contrastive loss landscape visualisation in Fig 3 is insightful; that the contrastive learning holds across datasets is quite convincing.

**Weaknesses:**

* **Robustness**: This work makes two main claims about the usefulness of their CL framework. Firstly, and convincingly so, about label efficiency. There are several places in the paper where model robustness is alluded to, or strong claims made, without any evidence. This could be because the authors view robustness simply to be good performance across multiple datasets? If so, this should be clarified. Currently, the claims about model robustness are misleading. See [1,2,3] for different robustness analyses of deep neural networks.

* **Assumption in Th. 3.2**: The guarantees of $Var[H_{SG}] < Var[H_{NS}]$ only holds if different $P_m$ do not have the same expected value over the aggregation function h(x;p). How is this ensured? In medical images, there are scenarios where the differences between classes are small in both intensity and feature spaces? How does the variance reduction guarantees hold in such situations?

* **Aggregation functions**: Aren't the aggregation functions some type of a distance measure? And why is it expensive to compute these on dense pixel grids (L213)?

* **Main method in Appendix**: While I appreciate all the details presented in this paper, moving the main method to the Appendix is not a good idea. Several of the important details are in the Appendix and does not serve the purpose of what Appendices are supposed to be.

* **Literature overview**: I was curious as to why the authors refrained from discussing self-supervised representation learning both when motivating the work, and also in their general discourse.

[1] Bastani, Osbert, et al. "Measuring neural net robustness with constraints." Advances in neural information processing systems 29 (2016).

[2] Carlini, Nicholas, and David Wagner. "Towards evaluating the robustness of neural networks." 2017 ieee symposium on security and privacy (sp). Ieee, 2017.

[3] Singh, Gagandeep, et al. "Fast and effective robustness certification." Advances in neural information processing systems 31 (2018).

**Questions:**

See Weaknesses.

**Limitations:**

Authors do not discuss any limitations of their work.

---

> ### Author Rebuttal · Authors · 2023-08-09
>
> We sincerely thank the reviewer for acknowledging our contribution to the self-supervised field, appreciating the strong performance improvement on multiple datasets, and providing constructive suggestions for the presentation of our work! We’ve made a substantial revision to the paper, which addresses all the issues, with emphasis on improving clarity of our work. If you have further concerns, please feel free to contact us.
>
> > **Q1**: clarifying robustness in our paper.
>
> **A1**: Thank you for the great suggestion! “Robustness” is a comprehensive concept used to describe the performance of a segmentation model. A model is said to be robust if (1) it has a high segmentation quality with only using extremely limited labels in long-tailed medical data (please see Line 141); (2) and fast convergence speed (please see Line 207). We agree with the reviewer and appreciate them for raising the concern about the wording. We will surely polish them in our final revision.
>
> > **Q2**: Assumption in Th. 3.2: The guarantees of variance reduction only holds if $P_m$  do not have the same expected value over the aggregation function h(x;p). How is this ensured? In medical images, there are scenarios where the differences between classes are small in both intensity and feature spaces? How does the variance reduction guarantees hold in such situations?
>
> **A2**: Thm 3.2 simply indicates that, provided that all the pixel groups $\{P_m\}_m$ do not have exactly same distributions, it is guaranteed $Var[\hat{H}(SG)] < Var[\hat{H}(NS)]$.
>
> Considering the unlikely condition where all pixel groups share an identical distribution within the feature space, it can be confidently asserted that SG will almost certainly realize a reduction in variance.
>
> Expanding on the previous point, the degree of variance reduction achievable, as anticipated, is contingent on the differentiation among pixel groups. We totally agree with you that there exist scenarios where the differences between classes are small in both intensity and feature spaces. Under these circumstances, the effect of variance reduction attributed to stratified group sampling would be comparatively insignificant. This condition aligns with the mathematical principle that variance is inherently linked to the extent of diversity in the dataset.
>
> > **Q3**: Aren't the aggregation functions some type of a distance measure? And why is it expensive to compute these on dense pixel grids (L213)?
>
> **A3**: Aggregation function refers to a function that is additive in pixels. That is, it can be expressed as a sum of functions of pixels, as is defined in Eq (3.1). Therefore, an aggregation function should be viewed as a general concept, which is not limited to a distance measure.
> In our paper, a loss function (e.g. $L_{contrast}$) is an example of an aggregation function. The computation would be overwhelming on dense pixel grids because in this case, we will have to aggregate information from all available pixel points across all images. Note that a typical 2D image can have 256x256 pixels, and there can be tens of thousands of training medical images (e.g., LiTS: 16684 slices). Thank you for the feedback, and we will further clarify in the revision.
>
>
> > **Q4**: While I appreciate all the details presented in this paper, moving the main method to the Appendix is not a good idea. Several of the important details are in the Appendix and does not serve the purpose of what Appendices are supposed to be?
>
> **A4**: Thank you for the great suggestion! We agree with your point that the relevant main method details should be in the main paper. In our final revision, we will follow your constructive advice to move the key details from the appendix to the main paper.
>
> > **Q5**: I was curious as to why the authors refrained from discussing self-supervised representation learning both when motivating the work, and also in their general discourse?
>
> **A5**: Thank you for your insightful suggestion! We agree with your view that an in-depth discussion on self-supervised representation learning would enhance the comprehension and context of our work. We truly appreciate your constructive advice and will be incorporating it to improve our manuscript. Given the limited space for this rebuttal, we aim to expand one subsection within the 'Related Work' section and offer a comprehensive analysis of self-supervised representation learning within the 'Introduction' section in our final revision. In line with your valuable advice, we're sharing a few snippets of our intended discussion below.
>  ```
> Self-supervised representation learning is a subclass of unsupervised learning, but with the critical distinction that it incorporates “inherent” supervision from the input data itself. The primary aim of self-supervised representation learning is to enable the model to learn the most useful representations from the large amount of unlabelled data for various downstream tasks. Self-supervised learning typically relies on pretext tasks, including predictive, contextual, and generative or reconstructive tasks. Among them, contrastive learning is considered as a popular approach for self-supervised representation learning by pulling the representations of similar instances closer and representations of dissimilar instances further apart in the learned feature space.
> ```
>
> Your advice is greatly appreciated. We will make corresponding revisions to clarify and provide details on them.
>
> Thank you once more for your valuable feedback! We have performed a substantial revision to the paper, placing particular emphasis on enhancing the clarity of our methodology. These changes will be incorporated into our final submission. We trust that these modifications better position our work for publication. Please do not hesitate to reach out if you have further comments or queries.

---

> > ### Comment · Reviewer_SMeb · 2023-08-16
> > **Response to author rebuttal**
> >
> > I have now read the author rebuttal, which addresses most of the concerns raised in my initial review. I have also seen the discussions between other reviewers and the effort invested by authors in these discussions is appreciated. I am willing raise my score from 6 t o7.

---

> > > ### Author Response · Authors · 2023-08-16
> > > **Thank you again for your review and very valuable feedback!**
> > >
> > > We sincerely thank you for taking the time to review our rebuttal and for recognizing the positive changes we have made to address the concerns raised. We deeply appreciate your consideration in adjusting the score. Your feedback has been invaluable in refining our work, and we are committed to ensuring the highest quality in our final manuscript. Again, we express our gratitude for your thoughtful review and positive reassessment!

---

### Official Review · Reviewer_LvuQ · 2023-07-01

**Soundness:** 4 excellent
**Presentation:** 4 excellent
**Contribution:** 4 excellent
**Rating:** 7
**Confidence:** 5

**Summary:**

The authors present ARCO, a novel semi-supervised contrastive learning framework that employs a stratified group sampling strategy (i.e. **SG** and **SAG**) to compute gradient estimators with reduced variance, thereby enhancing representation learning in dense contrastive learning. By improving dense contrastive learning, ARCO addresses issues related to class imbalancedness and enhances the performance of semi-supervised segmentation, particularly in scenarios with long-tail distributed anatomical classes. The authors provide theoretical evidence demonstrating the effectiveness of the proposed sampling techniques in reducing the variance of the aggregation function, specifically the contrastive loss. The efficacy of the proposed sampling technique is validated on eight 2D/3D benchmark datasets with different label settings, further reinforcing its effectiveness and practical applicability.

**Strengths:**


The motivations are clear, and the method is reasonable. This study shows me a new insight/direction to consider medical image segmentation. The following are my detailed comments.

---

**Empirical contribution**

Pixel/voxel-wise sampling constitutes a critical facet of contrastive learning at the pixel/voxel level. With the aid of variance-reduction estimation, the authors proffer two pragmatic approaches - Stratified Group (SG) and Stratified-Antithetic Group (SAG), tailored for pixel/voxel-level segmentation tasks with exceedingly scarce labels.

* The authors introduce a novel framework termed ARCO (strAtifed gRoup COntrastive learning) devised for multi-class segmentation tasks. This framework appears to be both intriguing and efficacious. The authors undertake a rigorous validation of the proposed methodologies across eight benchmark datasets, encompassing three 2D medical image segmentation, two 3D medical image segmentation, and three semantic segmentation benchmarks.

* The empirical results, both quantitative and qualitative, attest to the efficacy of the proposed model across all label ratios and datasets. For instance, the model demonstrates a marked enhancement in segmentation accuracy (up to 4.1% absolute improvements in Dice coefficient) on the challenging multi-class MMWHS dataset under a 1% label setting.

* The authors conduct comprehensive ablation studies to substantiate that the proposed mechanisms merit consideration. These studies encompass eight benchmark datasets, diverse network architectures, and varying label ratios to validate the efficacy, model-agnostic nature, and label efficiency of the proposed methodology.

* Lastly, the proposed methodologies are not only facile to implement but also boast of universal applicability. For instance, they can be seamlessly integrated into any scenario necessitating pixel/voxel sampling.  The paper presents a robust and versatile framework for pixel/voxel-level contrastive learning, which is empirically validated through extensive experiments and ablation studies. The methodologies are characterized by ease of implementation and broad applicability, making them a valuable contribution to the field of image segmentation.

---
**Theoretical contribution**

* The proposed methodologies, SG (Stratified Group) and SAG (Stratified-Antithetic Group), have exhibited remarkable efficacy in the experimental study. Consequently, it is intriguing to ascertain whether theoretical insights can elucidate this enhancement in performance. To this end, the paper furnishes a cogent theoretical analysis of the methodologies, revealing that the variance-reduction attribute of the two sampling methods is instrumental to their performance.

* First and foremost, in Section 3.3, the paper meticulously delineates the SG and SAG sampling methodologies through lucid mathematical equations. SG is executed by segregating pixels into mutually exclusive groups, followed by uniform sampling of a specified number of pixels from each group. SAG, which is predicated on SG, imposes an additional constraint of symmetry among the sampled pixels within each group.

* Subsequently, the sampled pixels are amalgamated through an aggregation function, which acts as an estimator for the target quantity. In the realm of image segmentation, this quantity could be, for instance, the contrastive loss function. It is posited that an optimal balance must be struck in the sample size; an overly diminutive sample size may fail to encapsulate the salient information from the underlying image, whereas an excessively large sample size would entail high computational complexity. Thus, the ideal scenario would be for SG to capture the crux of the image information through a relatively modest sample size. The paper demonstrates that SG possesses this attribute by establishing that it achieves reduced variance in comparison to the naïve sampling method (i.e., uniform random sampling from all pixels). Specifically, Theorem 3.2 establishes that SG is an unbiased sampling methodology, with a variance that does not exceed that of naïve sampling. More precisely, the variance of SG can be decomposed into the variance of naïve sampling minus a non-negative term. This non-negative term is conjectured to be almost certainly greater than zero, as it would be zero only if the expectation of the aggregation function within each group is identical to the expectation over the entire image, which is highly improbable. Thus, Theorem 3.2 suggests that SG is likely to consistently outperform naïve sampling.

* Figure 5 reveals that SG/SAG exhibits marginally expedited training convergence compared to naïve sampling. This observation is theoretically substantiated in the concluding paragraph of Section 3, which is commendable.

* In my assessment, the theoretical analysis presented in the paper is inextricably linked to the empirical component and provides a persuasive rationale for the empirical performance augmentation of SG vis-à-vis naïve sampling, thereby bringing the empirical narrative full circle.
---

**To sum up**

**1. Clarity**

The manuscript is eloquently composed, proffering a lucid and cogent progression of information. The authors adeptly elucidate the procedural framework, facilitating the readers' comprehension of the proposed methodologies, namely, the two instance sampling methods - Stratified Group (SG) and Stratified-Antithetic Group (SAG). In Section 3.3, the SG and SAG sampling methods are meticulously delineated through precise mathematical formulations. The discourse within the Methodology section (Section 3) furnishes an exhaustive exposition of the innovations introduced by the study, adeptly accentuating the distinct contributions of the research to the scholarly domain.

**2. Novelty**

This work is the first work, empirically and theoritically, to validate the variance-reduction approach within the context of pixel/voxel-level contrastive learning for semi-supervised medical image segmentation, particularly in scenarios characterized by a paucity of labels.

**3. Experimental Comprehensiveness**

The authors undertake a comprehensive suite of experiments encompassing eight medical datasets, which include both 2D and 3D modalities, as well as semantic segmentation benchmarks. Additionally, a diverse array of contrastive learning frameworks and varying label ratios are employed to rigorously assess the efficacy, model-agnostic properties, and label efficiency of the proposed methodology. This extensive experimental evaluation substantiates the robustness and versatility of the technique in the domain of medical image segmentation.

**4. Theoretical Implication**

The authors furnish a cogent and meticulously articulated theoretical analysis of the proposed approach, elucidating the underlying principles with clarity and precision. This analytical exposition contributes to a deeper understanding of the methodology's foundations and its implications.






**Weaknesses:**

The proposed method is very interesting. There is no obvious weakness in the proposed ARCO. However, I do have a few questions. See the following section.


**Questions:**

* The proposed two sampling techniques are intriguing. I wonder if you could kindly provide insights on the applicability of these sampling techniques to different tasks. Such information would prove beneficial not only to the current domain but also to other related domains and scenarios. Understanding the potential applications of these techniques would undoubtedly contribute to the advancement of various fields.

* Lemma 3.1 demonstrates that the variance of SAG is at most twice of that of SG, which implies that the variance reduction magnitude of SAG and SG are roughly at the same level. Could the authors further elaborate on when SAG be preferred to SG?


**Limitations:**

The authors have addressed the potential broader impact in clinical scenarios.

---

> ### Author Rebuttal · Authors · 2023-08-09
>
> We sincerely thank the reviewer for acknowledging our contribution and providing suggestions for the presentation of our work! In particular, we agree that discussing the potential application and more detailed classification are necessary to position this work properly. If you have further concerns, please feel free to contact us.
>
> > **Q1**: The proposed two sampling techniques are intriguing. I wonder if you could kindly provide insights on the applicability of these sampling techniques to different tasks. Such information would prove beneficial not only to the current domain but also to other related domains and scenarios. Understanding the potential applications of these techniques would undoubtedly contribute to the advancement of various fields?
>
> **A1**: Thank you for the great suggestion! Our sampling techniques can provide pragmatic solutions for enhancing variance reduction, thereby fostering their application in a wide array of real-world applications and sectors. These include but are not limited to 3D rendering, augmented reality (AR), virtual reality (VR), trajectory prediction, and autonomous driving. We agree with the reviewer and appreciate them for the constructive suggestion. We will add the potential applications in our final revision.
>
> > **Q2**: Lemma 3.1 demonstrates that the variance of SAG is at most twice that of SG, which implies that the variance reduction magnitude of SAG and SG are roughly at the same level. Could the authors further elaborate on when SAG be preferred to SG?
>
> **A2**: Thank you for the great suggestion! SAG, compared with SG, only samples half the number of pixels, and the other half are chosen as the opposite with respect to the group center. This is described by the equation $c_m -p = p’ - c_m$ following Line 219. In other words, these two pixels, $p$ and $p’$, are opposite of each other w.r.t. $c_m$.
>
> In general, SAG will outperform SG if the value of the component function $h(\cdot)$ are negatively correlated for two pixels $p$ and $p’$ that are opposite of each other w.r.t. $c_m$. To demonstrate this more rigorously, we consider the variance of the aggregate function restricted to two pixels, which can be written as $Var [h(p)+h(p’)]$.
>
> For SG sampling, $p$ and $p’$ are independently sampled in the pixel group $m$, and therefore we have $Var [h(p)+h(p’)] = Var[h(p)] + Var[h(p’)] = 2\sigma_m^2$. In stark contrast, for SAG, since the choice of $p’$ depends on $p$, we have $Var [h(p)+h(p’)] = Var[h(p)] + Var[h(p’)] + 2 Cov[h(p), h(p’)] = 2 \sigma_m^2 + 2 Cov[h(p), h(p’)]$. This indicates that, when $Cov[h(p), h(p’)] < 0$, i.e., $h(p)$ and $h(p’)$ are negatively correlated, it holds that $Var [h(p)+h(p’)] < 2\sigma_m^2$. In such cases SAG would have smaller variance than SG. We will make corresponding revisions and provide detailed elaborations of them.
>
> Overall, thank you again for your suggestions and review! We believe that discussing the potential application and more detailed classification will greatly improve the paper. We will include all the modifications in our final revision. We hope that the revision puts our work in better shape for publication. Please feel free to contact us for further concerns.

---

> > ### Comment · Reviewer_LvuQ · 2023-08-14
> > **Response to Authors' rebuttal**
> >
> > Thank you for your responses to the questions I raised. The response was satisfactory and addressed my concerns. Besides that, I'd also like to discuss/highlight the following point.
> >
> >
> > -  The notable contribution of the proposed SG sampling method in improving the representation quality for pixel/voxel-level contrastive learning. The superiority of SG over na&iuml;ve sampling has been shown across various datasets in the draft. Nonetheless, I would appreciate further insight into the SAG method. From my observation, SAG does not surpass SG in most experiments, except in the specific context of the SUN RGB-D benchmark under 50-label setting. Yet, the performance of SAG remains comparable to SG. Consequently, I interpret the SAG method as an alternative to SG, designed to achieve comparable results but with a reduced sample size. Would this be a correct interpretation?

---

> > > ### Author Response · Authors · 2023-08-14
> > > **Response to Reviewer LvuQ**
> > >
> > > We sincerely thank the reviewer for acknowledging the positive changes we have made to the paper. If you have further concerns, please feel free to contact us.
> > >
> > > > **Q1**: The notable contribution of the proposed SG sampling method in improving the representation quality for pixel/voxel-level contrastive learning. The superiority of SG over naïve sampling has been shown across various datasets in the draft. Nonetheless, I would appreciate further insight into the SAG method. From my observation, SAG does not surpass SG in most experiments, except in the specific context of the SUN RGB-D benchmark under 50-label setting. Yet, the performance of SAG remains comparable to SG. Consequently, I interpret the SAG method as an alternative to SG, designed to achieve comparable results but with a reduced sample size. Would this be a correct interpretation?
> > >
> > > **A1**: We thank the reviewer for acknowledging the positive changes we have made to the paper. Yes, your interpretation is correct! SAG can halve the sample sizes compared to SG, while largely preserving SG's variance reduction property, offering enhanced theoretical efficiency.
> > >
> > > We thank the reviewer again for the constructive feedback which helps shape this revision! Please do not hesitate to reach out should you have any additional feedback or questions.

---

> > > > ### Comment · Reviewer_LvuQ · 2023-08-15
> > > > **Response to authors' reply**
> > > >
> > > > Thanks for your quick reply. That works for me.
> > > > I don't have any other questions and would like to keep my original score and vote for "accept" for this solid work.

---

> > > > > ### Author Response · Authors · 2023-08-16
> > > > > **Thank you again for your review and very valuable feedback!**
> > > > >
> > > > > We deeply appreciate the reviewer's recognition of the improvements made to our paper. Once again, we extend our gratitude for the invaluable feedback that has greatly contributed to refining our work!

---

### Official Review · Reviewer_QWqe · 2023-07-02

**Soundness:** 3 good
**Presentation:** 3 good
**Contribution:** 3 good
**Rating:** 5
**Confidence:** 3

**Summary:**

This paper proposes two new sampling strategies, SG and SAG, for contrastive learning in semi-supervised frameworks. Compared with randomly sampling pixels for contrastive learning, pixels are grouped into several subsets, and then pixels are sampled from each subset. The proposed method is proven to reduce the variance of sampled pixels. Solid experiments are conducted to support the above claim.

**Strengths:**

1. Solid experiments and good performance.
2. Simple yet effective method that benefits contrastive learning for semi-supervised frameworks.
2. Sound theoretical analysis.

**Weaknesses:**

1. It would be better to also compare NS with the proposed SG/SAG in natural image datasets.
2. What is the application scenario of SAG? SG seems to have better performance and stability than SAG.
3. Line 222-223, what is the meaning of "p is orthogonal to p'"?

**Questions:**

See weaknesses.

**Limitations:**

The authors have included a discussion on the potential negative societal impact.

---

> ### Author Rebuttal · Authors · 2023-08-09
>
> We sincerely thank the reviewer for acknowledging our contribution to the medical image analysis field, appreciating the good performance improvement on our multi-class medical segmentation task, and providing constructive suggestions for the presentation of our work! We’ve made a substantial revision to the paper, which addresses all the issues, with emphasis on experiments in natural image datasets and clarity of the explanation of our work. If you have further concerns, please feel free to contact us.
>
> > **Q1**: It would be better to also compare NS with the proposed SG/SAG in natural image datasets?
>
> **A1**: Thank you for the great suggestion! We agree with your point that it is better to compare NS with the proposed SG/SAG on natural image datasets. Indeed we have compared NS with the proposed SG/SAG on three natural image datasets in Appendix (i.e, Cityscapes [84], Pascal VOC 2012 [85], indoor scene segmentation dataset – SUN RGB-D [86]). All the experiments are conducted under the same experimental setting [87]. For your convenience, the following Table shows the comparison results on Cityscapes, Pascal VOC, and SUN RGB-D (Please see Appendix K line 806 - 824 for qualitative/visual results).
> | | | | Pascal VOC | | | CityScapes | | | | SUN RGB-D  | | |
> | :----------- | -----------: | :-----------: | :-----------: | :-----------: | -----------: | :-----------: | :-----------: | :-----------: | -----------: | :-----------: | :-----------: | :-----------: |
> | Method | 60 labels |120 labels | 600 labels | all labels |20 labels| 50 labels |150 labels | all labels | 50 labels |150 labels |500 labels|all labels|
> | Supervised | 39.4 | 55.5 | 64.6 | 77.8 | 38.2 | 45.9 | 55.4 | 70.9 | 20.0 | 29.2 | 38.9 | 51.8 |
> | ReCo [87] + ClassMix | 57.1 | 69.4 | 73.2 | - | 49.9 | 57.9 | 65.0 | - | 30.5 | 40.4 | 44.6 | - |
> | ARCO-SAG (9 Grid) + ClassMix | 58.3 | 70.5 |75.4 | - | 50.2 | 60.2 | 66.5 | - | 31.5 | 40.9 | 45.7 | - |
> | ARCO-SAG (16 Grid) + ClassMix | 58.7 | 70.9 | 75.1 | - | 50.1 | 60.6 | 66.3 | - | 37.8 | 40.2 |45.7 | - |
> | ARCO-SAG (25 Grid) + ClassMix | 59.1 | 70.9 | 74.9 | - | 49.8 |60.6 | 66.7 | - |**38.5**| 40.5 | 45.5 | - |
> | ARCO-SG (9 Grid) + ClassMix | 59.2|**71.8**| 75.3 | - | 52.5 | 60.9 | **66.8**| - | 32.4 | 41.4 | 46.6 | - |
> | ARCO-SG (16 Grid) + ClassMix |**59.6**| 71.7 |**75.5**| - |**53.7**|61.2| 66.2 | - | 37.7 |41.0 | 46.4 | - |
> | ARCO-SG (25 Grid) + ClassMix | 59.5|71.7| 75.2 | - | 51.5 |**61.8**|66.4| - |38.3|**41.5**|**47.3**| - |
>
> As we can see, we can see that for all cases, SG and SAG consistently improve performance, compared to NS, in all the semi-supervised settings. The results, quantitatively (Appendix Table 5 Page 25) and qualitatively (Appendix Page 26 - 34) clearly prove the effectiveness of our proposed SG/SAG. Due to the limited response space, we will surely provide detailed discussions on them in our revision.
>
> > **Q2**: What is the application scenario of SAG? SG seems to have better performance and stability than SAG.
>
> **A2**: We greatly appreciate your attention to this matter. SAG, compared with SG, only samples half the number of pixels, and the other half are chosen as the opposite with respect to the group center, making it theoretically more efficient. This is described by the equation $c_m -p = p’ - c_m$ following Line 219. In other words, these two pixels, $p$ and $p’$, are opposite of each other w.r.t. $c_m$. Consequently, this theoretical efficiency suggests a wider scope for SAG in various real-world applications, encompassing fields like 3D rendering, augmented reality (AR), and virtual reality (VR).
>
> To be more specific, SAG will outperform SG if the value of the component function $h(\cdot)$ are negatively correlated for two pixels $p$ and $p’$ that are opposite of each other w.r.t. $c_m$. To demonstrate this more rigorously, we consider the variance of the aggregate function restricted to two pixels, which can be written as $Var [h(p)+h(p’)]$.
>
> For SG sampling, $p$ and $p’$ are independently sampled in the pixel group $m$, and therefore we have $Var [h(p)+h(p’)] = Var[h(p)] + Var[h(p’)] = 2\sigma_m^2$. In stark contrast, for SAG, since the choice of $p’$ depends on $p$, we have $Var [h(p)+h(p’)] = Var[h(p)] + Var[h(p’)] + 2 Cov[h(p), h(p’)] = 2 \sigma_m^2 + 2 Cov[h(p), h(p’)]$. This indicates that, when $Cov[h(p), h(p’)] < 0$, i.e., $h(p)$ and $h(p’)$ are negatively correlated, it holds that $Var [h(p)+h(p’)] < 2\sigma_m^2$. In such cases SAG would have smaller variance than SG. Due to the limited response space, we will ensure a more detailed discussion on these methods and their potential applications in our final revision.
>
> > **Q3**: Line 222-223, what is the meaning of "p is orthogonal to p' "?
>
> **A2**: The “orthogonal” sign means ‘independent’. In line 222, "p is orthogonal to p' " means that the pixel p and pixel p’  are sampled independently. Thank you for the suggestion, and we will further clarify in the revision.
>
> We deeply appreciate your patience and engagement throughout this review process. We have conducted an extensive revision of our paper, which addresses all the issues. This revision focuses particularly on enhancing the clarity of our method's explanation and has conducted appropriate comparisons through experiments on three natural image datasets. We believe that these improvements have significantly shaped our work for the better and brought it closer to being ready for publication. Please do not hesitate to reach out to us should you have any further concerns. Your ongoing interest and insightful feedback on our work are greatly valued.

---

> > ### Comment · Area_Chair_Tp4X · 2023-08-18
> > **Rebuttal to QWqe**
> >
> > Dear QWqe
> >
> > Could you have a look at the rebuttal to see if your questions have been clarified?
> >
> > Thanks,
> > Your AC

---

> > ### Author Response · Authors · 2023-08-20
> > **Look forward to further feedback**
> >
> > Dear Reviewer QWqe:
> >
> > We are genuinely thankful for your thoughtful feedback, which has been pivotal in refining our manuscript.
> >
> > As the author-reviewer discussion period is nearing its conclusion, we kindly request your review of our rebuttal and any further reflections you might have. Please feel free to indicate any additional clarifications or experiments that could further strengthen our paper. We aim to unequivocally convey the significance of our work. If you feel that our responses have adequately addressed your concerns, we would be grateful if you might consider raising the paper's rating.
> >
> > Once again, we deeply value your comprehensive review and are thankful for the positive evaluation. Your feedback has been pivotal in refining our work, and we are committed to incorporating all of the suggestions into our manuscript.
> >
> > Best
> >
> > Authors of Paper1499

---

### Official Review · Reviewer_J4c5 · 2023-07-04

**Soundness:** 3 good
**Presentation:** 1 poor
**Contribution:** 2 fair
**Rating:** 4
**Confidence:** 4

**Summary:**

The authors propose a sampling strategy to improve contrastive-learning-based medical image segmentation performance with limited labeled data and training stability. The sampling strategy is used in conjunction with a previously-published contrastive semi-supervised training strategy. The main contributions include this sampling strategy, some theoretical support that the strategy should improve training stability, and experiments evaluating the method.

**Strengths:**

- The baseline comparison experiments are thorough.
- The authors address a real applied problem (that most medical image segmentation datasets have limited data) with a new method and theoretical support, resulting in a well-rounded paper.


**Weaknesses:**

Major weaknesses
- The paper’s writing is a major limitation. The prose is difficult to understand, which limits the entire paper—it is hard to clearly understand the motivation, the proposed method, the contributions, or the benefits. The work would benefit from more rounds of grammatical revision; it is hard to parse what the author is trying to communicate a lot of the time.
- Partly due to the writing issues, it is unclear how the proposed sampling strategy differs from previous dense contrastive learning-based image segmentation strategies; it seems that the methodological contributions here are minor, if existent.
- The method is complex, consisting of two backbones (connected via EMA), global/local instance discrimination losses, augmentations, supervised losses, nearest neighbor loss, global contrastive loss, unsupervised loss… with all of these components, a very thorough ablation section is needed to understand how much the novel component (a pixel-level sampling strategy) is contributing to performance. The existing ablation section is not so thorough. As a result, the paper does not contribute much understanding about the strengths/weaknesses of different components of this complex pipeline.

Minor weaknesses
- MONA is not a well-known training strategy; it would be useful to provide a longer overview on what MONA does and how the proposed approach differs. This discussion could go in an appendix.
- The same concept is often referred to using different words: model “convergence,” “robustness,” and “stability.” I’m not sure if you’re always talking about the same concept, or if you are using different words to refer to the same idea. If the latter, it helps the reader to always use the same word.
- This is a minor stylistic note, but the use of bold and italics is often distracting.


**Questions:**

Questions and suggestions discussed above

**Limitations:**

Limitations and impacts adequately addressed.

---

> ### Author Rebuttal · Authors · 2023-08-09
>
> We sincerely thank the reviewer for acknowledging our contribution, appreciating the strong performance improvement on our medical image segmentation tasks, and providing constructive suggestions for the presentation of our work! We’ve made a substantial revision to the paper, which addresses all the issues, with emphasis on clarity of the explanation of our work and appropriate ablation experiments on different components. If you have further concerns, please feel free to contact us.
>
> > **Q1**: Writing revision?
>
> **A1**: Thank you for the great suggestion! We will thoroughly assess the wording in our current manuscript and polish them accordingly. Here we highlight our motivation and contribution.
> Our motivation lies in the observation that the sampling procedure in contrastive learning introduces an additional source of variance, which could result in model collapse and undermine the overall performance of the model.
> To this end: we have devised two sampling techniques that (1) are easy to implement, functioning in a plug-and-play manner; (2) theoretically demonstrate variance reduction properties; and (3) empirically deliver improved segmentation quality in 8 benchmarks, i.e., 5 2D/3D medical and 3 semantic segmentation datasets, with different label settings.
>
> > **Q2**: Difference between our sampling strategy and dense contrastive learning (CL) segmentation.
>
> **A2**: Existing dense CL segmentation methods [46,20] use patch-level strategy, e.g., [46] first partitions the image into fixed-size patches (3x3), and then randomly selects 13 patches per image. For each of these chosen patches, they further select 5 negative samples from patches located in different regions of the same image. Thus, they implement dense CL with these local regions, as in Eqn. 3 [46]. In contrast, our SG/SAG first partition the image with respect to different classes into grids with the same size, and then sample, within the same grid, pixels semantically close to each other with high probability. Thus, we implement CL with these sampled different class pixels, as in Eqn. E2 (Appendix Line 766 - 767). This essentially allows us to obtain better segmentation quality and label efficiency by improving feature variance reduction in training, as shown in Sections 4.
>
> > **Q3**: Ablation on different components.
>
> **A3**: Thank you for your constructive feedback! We agree that the importance of providing a comprehensive ablation study to discern the contribution of each individual component, especially the novel pixel-level sampling strategy (Please see **“global” response PDF file** for requested ablations).
>
> To clarify, our work focuses on semi-supervised medical image segmentation, where we adhere to the widely accepted training standards, such as supervised loss, data augmentation, and EMA training, as referenced in [27,52,5,81,49]. Furthermore, to substantiate the necessity of the various components, we refer to the one-page **“global” response PDF file**. Tables 1,2,3 show the comparative results of various components, including pixel-level sampling variants (i.e., naive sampling (NS), Stratified Group (SG), and Stratified-Antithetic Group (SAG)), contrastive loss, nearest neighbor loss, unsupervised loss, and the global/local instance discrimination losses on the ACDC dataset with a 1% label ratio. These findings demonstrate:
>
> 1. The positive contribution of each component to performance gains.
> 2. The benefits of employing SG/SAG sampling over the NS setting, resulting in marked improvements in Dice.
> 3.  Figure 5 (Line 323 - 347) illustrates how SG/SAG enhance convergence with reduced standard deviations, indicating enhanced robustness.
>
> Our results underscore the efficacy of our proposed method, especially in the realm of medical image segmentation. We are grateful for your valuable suggestions and will ensure these findings and clarifications are prominently featured in our final revision. We remain receptive to any further insights to enhance the quality of our paper.
>
> > **Q4**: Overview MONA.
>
> **A4**: Thank you for pointing this out! Indeed we have conducted a comprehensive review of the MONA framework. Due to limited rebuttal space, please refer to Appendix C (Lines 691 - 716), E (Lines 732 - 777), and F (Lines 778 - 789) for further details. In our final revision, we will follow your constructive advice to transfer the key details from the Appendix to the main paper for better clarity and visibility.
>
> > **Q5**: model convergence, robustness, and stability are unclear.
>
> **A5**: Many thanks! Here we further clarify the terms “convergence,” “robustness,” and “stability” and the difference between them.
> Specifically, “convergence” is a common concept which simply refers to the training speed, which is the number of epochs required to learn an accurate model.
> “Stability” is a desired property of a sampling method. It refers to the standard deviation of the sampling method. A stronger stability means the sampling method has a small standard deviation, which is good in the pixel sampling procedure of contrastive learning.
> “Robustness” is a comprehensive concept used to describe the performance of a segmentation model. A model is said to be robust if (1) it has a high segmentation quality with only using extremely limited labels in long-tailed medical data (please see Line 141); (2) and fast convergence speed (please see Line 207).
>
> > **Q6**: This is a minor stylistic note, but the use of bold and italics is often distracting.
>
> **A6**:  Thanks for pointing this out. We will modify it accordingly.
>
> We have extensively revised our manuscript, focusing on enhancing the clarity of our explanations and conducting detailed ablation studies on various components. We trust these adjustments improve the paper's readiness for publication. Please do not hesitate to reach out should you have any additional feedback or questions.

---

> > ### Comment · Reviewer_J4c5 · 2023-08-11
> > **Rebuttal response**
> >
> > I have read the other reviews and rebuttal responses and I appreciate the authors’ thorough rebuttal. I have reread the paper as well during this rebuttal period. To summarize my thoughts:
> > - I think the paper shows promising empirical results at training segmentation models with few labeled images. I am encouraged by the authors’ addition of of new ablation experiments.
> > - I think reviewer SMeb brings up a good point about the term and associated claims of “robustness” that I did not include in my original review.
> >  - I still find the paper very difficult to interpret and the proposed training method to contain so many components that it is difficult to know which components are useful for other pipelines.
> >
> > I recognize that the other reviewers do not seem to have the same perspective. I will change my score from a 3 to a 4 due to the additional experimental support during this rebuttal period, but I would still vote to reject this submission. I would point the AC to Sections 3.2 and 3.3 of the paper, which contain the paper’s contributions, if they wish to investigate further and see if they find the same problems I do with the submission or if they are more aligned with the other reviewers.

---

> > > ### Author Response · Authors · 2023-08-13
> > > **Response to Reviewer J4c5 (Part 1)**
> > >
> > > Thank you for taking the time to re-examine our paper during the rebuttal phase! We'd like to further address your additional queries as follows:
> > >
> > > > **Q1**: I think the paper shows promising empirical results at training segmentation models with few labeled images. I am encouraged by the authors’ addition of of new ablation experiments.
> > >
> > > **A1**: We appreciate your recognition of the paper's promising empirical results and our commitment to improving the manuscript with the addition of new ablation experiments. If you have further concerns on ablation studies, please feel free to contact us.
> > >
> > > > **Q2**: I think reviewer SMeb brings up a good point about the term and associated claims of “robustness” that I did not include in my original review.
> > >
> > > **A2**: Many thanks! We appreciate the opportunity to provide clarity on the term "robustness": In this context, “robustness” is a comprehensive concept used to describe the performance of a segmentation model. A model is said to be robust if (1) it has a high segmentation quality with only using extremely limited labels in long-tailed medical data (please see Line 141); (2) and fast convergence speed (please see Line 207).
> > >
> > > Our introduced sampling methods, namely Stratified Group (SG) and Stratified-Antithetic Group (SAG), have shown improvements in segmentation performance across different pipelines. Furthermore, these methods can seamlessly integrate with existing frameworks, further enhancing the robustness (as detailed in Appendix Lines 825-840, Section L and Appendix Page 25, Table 6).
> > >
> > > We greatly appreciate the feedback and will make it a priority to refine and highlight this to improve the paper's clarity.

---

> > > > ### Author Response · Authors · 2023-08-13
> > > > **Response to Reviewer J4c5 (Part 2)**
> > > >
> > > > Thanks for your helpful comments! If you have further concerns, please feel free to contact us.
> > > >
> > > > > **Q3**: I still find the paper very difficult to interpret and the proposed training method contains so many components that it is difficult to know which components are useful for other pipelines. I would point the AC to Sections 3.2 and 3.3 of the paper, which contain the paper’s contributions.
> > > >
> > > > **A3**: We sincerely value your feedback and will thoroughly revise the section in our final revision. We first clarify that we do **NOT** claim the training pipelines as our contribution. Instead, we have introduced two plug-and-play sampling techniques that seamlessly integrate with existing state-of-the-art Contrastive Learning (CL) frameworks adopting the student-teacher design. Section 3.2 discusses the potential high variance in CL frameworks and presents strategies for addressing these challenges along with their implementation details. Section 3.3 offers a theoretical analysis explaining the efficacy of our proposed method in enhancing variance reduction, accompanied by optimization guarantees (detailed further in Appendix, pages 595-647, Sec. A). Below, we would like to offer further clarification on our motivation and contributions.
> > > >
> > > > #### **Major Motivations**:
> > > >
> > > > Intuitively, using Contrastive Learning (CL) aids in learning robust representations for medical segmentation if positive and negative pairs align with the intended latent anatomical classes [46,13]. However, recent studies suggest potential issues, such as representation collapse into constant features [11,10,16] or confinement to a lower-dimensional subspace [17,18,19,20]. A key contributor to this fragility might be the non-smooth feature space near samples [21,22], where sampling of pixels could introduce significant feature variations. Existing sampling methods, such as naive sampling, often produce higher variances and struggle to discern semantically similar pixels [21], potentially compromising CL's stability. This prompts the pivotal query: How might we adeptly select the most informative pixels/voxels, thereby reducing variance in semi-supervised CL models? As illustrated in Figure 3 (Lines 217-218), regions with similar anatomical features within medical images should cluster, yielding corresponding plateaus in the loss landscape visualization, a perspective echoed in recent empirical studies [66,67].
> > > >
> > > > #### **Key Contributions**:
> > > > - **Technical Novelties**: We introduce two sampling techniques, i.e., Stratified Group (SG) and Stratified-Antithetic Group (SAG), to mitigate the undesirable high-variance limitation through the concept of variance-reduced estimation.
> > > > - **Robustness Across Different Pipelines**: Through extensive experimental results, we demonstrate that two proposed sampling methods combined with state-of-the-art CL frameworks perform well in pixel/voxel-level segmentation tasks with extremely limited labels (further detailed in Appendix Lines 825-840, Section L, and Appendix Page 25, Table 6).
> > > > -  **Theoretical Analysis**: To our best knowledge, we are the **first work** to show the benefit of certain variance-reduction techniques in CL for medical image segmentation. We unveil the untapped potential of the refined gradient estimator for handling long-tailed medical image data.
> > > > - **Comprehensive Experiments**: We conducted extensive experiments on **eight** diverse 2D/3D medical and semantic segmentation benchmarks, demonstrating the efficacy of our approach across various label ratios, i.e., 1\%, 5\%, 10\% (further detailed in Lines 261-319 Section 4.1, Line 247-248 Table 1, Line 273-274 Figure 1, and Appendix Page 22-35, Sections G-K).
> > > >
> > > > Thank you for your continued consideration and time! Please feel free to contact us for further concerns.

---

> > > > > ### Author Response · Authors · 2023-08-21
> > > > > **Look forward to further feedback**
> > > > >
> > > > > Dear Reviewer J4c5:
> > > > >
> > > > > As the review period draws to a close, we would like to emphasize our gratitude for your insightful comments, which have significantly shaped our revisions. We wish we have successfully addressed your concerns.
> > > > >
> > > > > We sincerely value your feedback and insights and humbly request your endorsement. We wish to emphasize our meaningful contribution to the realm of semi-supervised medical image segmentation. Our work also provides the community with significant insights into generic vision intelligence, especially in understanding the nuances of current self-supervision objectives for complex, safety-critical tasks. We believe our research can substantially enrich the collective knowledge in this domain.
> > > > >
> > > > > With a brief window remaining for discussion, we remain open to further dialogue to address any residual questions or concerns. Your continued engagement is highly regarded and vital to our research aims. Thank you once again for your dedication and thoughtful consideration.
> > > > >
> > > > > Best
> > > > >
> > > > > Authors of Paper1499

---

### Author Rebuttal · Authors · 2023-08-09

In response to the query from Reviewer-J4c5 (Question 3), we appreciate the emphasis on the depth of our analysis. Given the constraints of this rebuttal format, we wish to assure our commitment to augmenting the ablation section for a more comprehensive understanding.

To fully understand the contribution of each component, especially our novel pixel-level sampling strategy, we present comparative outcomes of components. These include pixel-level sampling variants (i.e., naive sampling (NS), Stratified Group (SG), and Stratified-Antithetic Group (SAG)), global contrastive loss, nearest neighbor loss, unsupervised loss, the global/local instance discrimination losses, and data augmentations assessed on the ACDC dataset with a 1% label ratio. For a detailed overview, please consult the “global” response PDF file.

---

### Decision · Program_Chairs · 2023-09-21

**Decision:**

Accept (poster)

**Comment:**

The paper introduces a couple of sampling strategies to improve contrastive learning in semi-supervised learning for medical image segmentation. The proposed method is theoretically shown to reduce the variance of sampled pixels, which improves training stability. Experiments show empirical evidence for claims. The paper received scores 7,7,5,4, where the main strengths are: 1) thorough experiments, 2) realistic problem, and 3) sound theoretical analysis. The main weaknesses are: 1) unclear writing; 2) method complexity; 3) missing comparison in natural image datasets; and 4) unclear claims. The rebuttal seems to have addressed many of the issues. There are some good suggestions to improve paper clarity, so I highly encourage the authors to take the suggestions into consideration for the preparation of the final version of the paper.